# Evolved bacterial resistance against fluoropyrimidines can lower chemotherapy impact in the *Caenorhabditis elegans* host

Brittany Rosener[1], Serkan Sayin[1], Peter O Oluoch[1], Aurian P García González[1], Hirotada Mori[2], Albertha JM Walhout[1,3], Amir Mitchell[1,3,4]*

[1]Program in Systems Biology, University of Massachusetts Medical School, Worcester, United States; [2]Data Science Center, Nara Institute of Science and Technology, Ikoma, Japan; [3]Program in Molecular Medicine, University of Massachusetts Medical School, Worcester, United States; [4]Department of Molecular, Cell and Cancer Biology, University of Massachusetts Medical School, Worcester, United States

**Abstract** Metabolism of host-targeted drugs by the microbiome can substantially impact host treatment success. However, since many host-targeted drugs inadvertently hamper microbiome growth, repeated drug administration can lead to microbiome evolutionary adaptation. We tested if evolved bacterial resistance against host-targeted drugs alters their drug metabolism and impacts host treatment success. We used a model system of *Caenorhabditis elegans*, its bacterial diet, and two fluoropyrimidine chemotherapies. Genetic screens revealed that most of loss-of-function resistance mutations in *Escherichia coli* also reduced drug toxicity in the host. We found that resistance rapidly emerged in *E. coli* under natural selection and converged to a handful of resistance mechanisms. Surprisingly, we discovered that nutrient availability during bacterial evolution dictated the dietary effect on the host – only bacteria evolving in nutrient-poor media reduced host drug toxicity. Our work suggests that bacteria can rapidly adapt to host-targeted drugs and by doing so may also impact the host.

*For correspondence: amir.mitchell@umassmed.edu

Competing interests: The authors declare that no competing interests exist.

## Introduction

The microbiome plays a critical role in disease progression and can influence treatment success through a myriad of direct and indirect interactions with its host. Metabolism of host-targeted drugs by commensal bacteria emerges as a major mechanism underlying some of these complex interactions with clear clinical implications in cancer, HIV, and additional diseases (*Pryor et al., 2019a*; *Spanogiannopoulos et al., 2016*). Recent systematic screens revealed that bacterial metabolism substantially alters 65% of host-targeting drugs (*Lehouritis et al., 2015*; *Zimmermann et al., 2019*) and in vivo studies demonstrated that bacterial metabolism has clinical implications in multiple cases (*Spanogiannopoulos et al., 2016*). Interestingly, it has also become apparent that the interactions between the microbiome and host-targeted drugs are bidirectional; while drugs are frequently metabolized by the microbiome, the microbiome itself is also often inadvertently harmed by host-targeted drugs. A recent screen found that 25% of host-targeted drugs considerably hamper the growth of representative species of the gut microbiome at physiological concentrations (*Maier et al., 2018*). The forces shaping microbiome–drug–host interactions are therefore highly intertwined – host-targeted drugs apply selective pressure on the microbiome which leads to

microbiome adaptation. A key question that emerges is whether microbiome adaptation to host-targeting drugs will in turn influence bacterial drug metabolism and therefore impact the treated host.

Microbiome changes take place through two fundamental processes: shifts in species composition (ecological changes) and fixation of beneficial mutations within individual species (intraspecies adaptation) (*Garud et al., 2019*). Multiple metagenomic studies using 16S rRNA gene sequencing have repeatedly identified shifts in species composition in response to changes in host diet and drug treatment (*Forslund et al., 2015*; *Imhann et al., 2016*; *Jackson et al., 2018*; *Vich Vila et al., 2020*). Concordantly, recent works also uncovered microbiome changes through fixation of spontaneous adaptive mutations in individual species (*Lourenço et al., 2016*; *Sousa et al., 2017*; *Zhao et al., 2019*). While the study of intraspecies adaptation still lags behind the study of species composition, pioneering studies have established that intraspecies adaptation is a key component in microbiome adaptation. Longitudinal studies in mouse models showed that the gut environment rapidly selects for spontaneously arising beneficial mutations that can reach fixation within a few days (*Barroso-Batista et al., 2014*; *Crook et al., 2019*; *Lourenço et al., 2016*). Longitudinal studies in humans similarly uncovered evidence of adaptive intraspecies evolution in the digestive tract (*Crook et al., 2019*; *Garud et al., 2019*; *Zhao et al., 2019*).

Over the years, model organisms such as *Caenorhabditis elegans*, Drosophila, and zebrafish have emerged as useful tools for disentangling the complexity of microbiome–drug–host interactions (*Douglas, 2019*; *Maier and Typas, 2017*). Specifically, *C. elegans* and its bacterial diet have been used as a model system for the discovery and mechanistic understanding of bacterial influences on the efficacy of drugs commonly prescribed to treat cancer and diabetes (*García-González et al., 2017*; *Pryor et al., 2019b*; *Scott et al., 2017*). Studies on antimetabolites, a large class of drugs that are widely used to treat cancer and viral infection, are of special interest since evidence indicates that microbiome metabolism can significantly reduce their efficacy in vivo (*Geller et al., 2017*; *Klatt et al., 2017*; *Lehouritis et al., 2015*). The two closely related pro-drugs 5-fluorouracil (5-FU) and 5-fluoro-2'-deoxyuridine (FUDR) are two examples of such antimetabolites that are used as chemotherapy. The pro-drugs, masquerading as extracellular nucleotides, are transported into the cell and are sequentially modified into toxic compounds by the nucleotide synthesis network. These antimetabolites are also potent antimicrobial agents since their toxicity originates from a key metabolic pathway found in all cell-types – nucleotide synthesis. In bacteria, intermediate compounds metabolized from these fluoropyrimidines interfere with DNA and RNA synthesis, inhibit the synthesis of deoxythymidine monophosphate (*Cohen et al., 1958*), and lead to harmful accumulation of cell wall precursors (*Tomasz and Borek, 1962*; *Tomasz and Borek, 1960*). Recent studies on these drugs revealed that mutations in the bacterial gene network for nucleotide synthesis can both increase and decrease drug efficacy in the *C. elegans* host feeding on mutated bacteria (*Figure 1A*). The main mechanism underlying this host effect is bacterial accumulation of the pro-drug intermediate 5-fluorouridine 5'-monophosphate (FUMP) that is highly cytotoxic for the animal (*García-González et al., 2017*; *Scott et al., 2017*).

The mechanistic insights provided by previous studies in the *C. elegans* model system allowed us to address an unexplored open question: what are the consequences of microbiome adaptation to host-targeted drugs on the host itself? Here we used *C. elegans* fed by *Escherichia coli* and exposed to two fluoropyrimidine drugs as a model system to investigate if bacterial adaptation will impact the host. We show that in this model system host-impact is frequently observed and that bacterial adaptation can take place rapidly since it leverages gene inactivation. Our study consists of three parts (*Figure 1B*): a genetic screen to systematically map resistance mechanisms in bacteria, an in vitro evolution experiment to monitor naturally evolving bacterial drug resistance over short evolutionary time scales and its host implications, and an investigation of the molecular mechanisms underlying evolved bacterial drug resistance. The genetic screen revealed that the bacterial 'Resistome', the set of genes conferring resistance upon inactivation, is highly overlapping for the two drugs and includes multiple pathways that were not previously associated with resistance. Interestingly, we discovered that the Resistome greatly differs when the screens were conducted in nutrient-poor and nutrient-rich growth media. The screen, therefore, indicated that resistance can emerge rapidly (through inactivation of a single gene) and that the resistance mechanism would depend on the extracellular conditions during selection. The results from our in vitro evolution confirmed this premise and revealed that the driver mutations underlying drug resistance were indeed nutrient dependent. Remarkably, we also discovered that although all evolved bacteria become drug

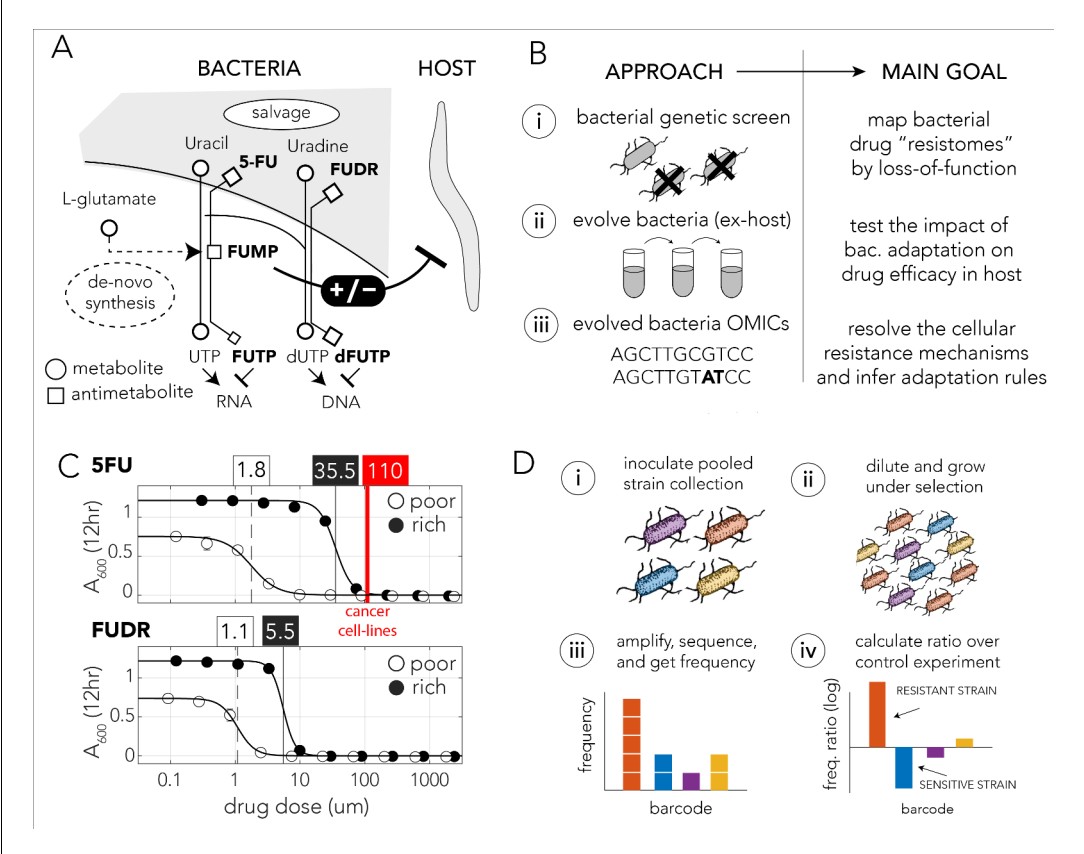

**Figure 1.** Bacteria effect on host drug toxicity and study design. (**A**) Toxicity mechanisms of 5-FU and FUDR pro-drugs in bacteria and *Caenorhabditis elegans*. The two molecules, masqueraded as pyrimidines, are transported into bacteria through the nucleotide salvage pathway and are metabolized into active compounds that interfere with RNA and DNA synthesis. The intermediate derivative FUMP that is produced by the bacteria is highly toxic for the *C. elegans* host. Bacterial mutations effecting FUMP metabolism can increase or decrease drug toxicity in a *C. elegans* that is feeding on the bacteria and is exposed to the drug. (**B**) Overview of the study's approach and aims of individual study stages. (**C**) Dose–response curves of 5-FU and FUDR in *Escherichia coli* and the calculated $IC_{50}$ (half maximal inhibitory concentration). Both drugs are considerably more toxic for bacteria in nutrient-poor media. The mean $IC_{50}$ of 5-FU from 806 human cancer (red line) is higher than the $IC_{50}$ measured for *E. coli*. (**D**) Overall approach for pooled screening with the *E. coli* barcoded strain collection. The frequency of individual barcodes can be measured by deep sequencing of the barcode locus and can be used to infer changes in barcode representation in different conditions. The relative frequency of a barcode in screen and control experiments was used to infer if the gene knockout corresponding to the barcode increases drug resistance.

The online version of this article includes the following figure supplement(s) for figure 1:

**Figure supplement 1.** Sequencing statistics of pooled genetic screens.
**Figure supplement 2.** Volcano plots for the identification of screen hits.
**Figure supplement 3.** Drug resistance of individual knockout strains.
**Figure supplement 4.** Network of related GO categories that are enriched by the genetic screen.

resistant, only strains that grew in nutrient-poor media selected for resistance mutations that reduced drug toxicity in the host. Taken together, our results provide a proof-of-concept for the premise that adaptation of individual species within the microbiome to the two tested drugs can impact the efficacy of host-targeted drugs and suggests that evolution converges to predictable resistance mechanisms that depend on nutrient availability.

## Results

### Bacterial 5-FU and FUDR resistomes

We first characterized the inhibitory concentrations of 5-FU and FUDR in *E. coli* by monitoring growth inhibition in different drug concentrations after 12 hr of growth. We predicted that

antimetabolite toxicity will be reduced if the growth media contains competing metabolites, such as nucleobases, nucleosides, and nucleotides and therefore repeated the measurement in nutrient-poor and nutrient-rich media. *Figure 1C* shows the observed drug sensitivity curve. Calculation of the $IC_{50}$, the concentration of a drug that gives half-maximal growth inhibition, showed that drug sensitivity is media dependent: bacteria growing on nutrient-poor media that lacks potentially competing metabolites were significantly more sensitive to both drugs. The $IC_{50}$ for 5-FU was 1.8 μM and 35.5 μM in nutrient-poor and nutrient-rich media, respectively. The $IC_{50}$ for FUDR was 1.1 μM and 5.5 μM in nutrient-poor and nutrient-rich media, respectively. Interestingly, systematic measurements across 806 human cancer cell lines revealed a higher average inhibitory concentration of 110 μM 5-FU (*Yang et al., 2013*). Thus, *E. coli* is more drug sensitive than many cell types tested from the human host.

We next performed a genetic screen to identify loss-of-function mutations that increase resistance against the 5-FU and FUDR in nutrient-poor or nutrient-rich media. In these screens we used a pooled collection of 3680 single-gene knockout strains. This strain collection is similar to the widely used Keio collection of single-gene deletion strains (*Baba et al., 2006*), yet in this library deleted genes are replaced by a resistance cassette and a unique barcode sequence of 20 bp. This cloning strategy allowed us to infer drug sensitivity of all knockout strains in a pooled screen by amplifying and sequencing the barcode region. A similar approach has been widely applied in *Saccharomyces cerevisiae* (*Giaever et al., 2002*; *Giaever and Nislow, 2014*). Using this library, we inferred the sensitivity of each knockout strain to either drug by measuring the relative frequency of the corresponding barcode in a screen condition and calculating its ratio over the relative frequency of the same barcode in a no-drug control experiment. A ratio significantly higher than one indicated that the barcode was over-represented in the drug screen and therefore that the corresponding gene knockout conferred resistance (*Figure 1D*).

We identified an average of 1.5 million independent barcodes in each screen with an average coverage of about 500 reads for each unique barcode (*Figure 1—figure supplement 1A*). We identified a perfect match to a known barcode in 85% of sequenced reads. Reassuringly, we observed a very high correlation (Pearson r > 0.93) when we compared the frequency of barcodes in biological duplicates (*Figure 1—figure supplement 1B*). This observation indicated that replicates capture a highly similar pattern of resistant strains that likely corresponds to the selective pressure of the drug. Interestingly, a pairwise correlation between all screen conditions revealed a hierarchical structure of similarity (*Figure 1—figure supplement 1C*), with biological duplicates being most correlated, followed by a clear bifurcation by media type. This correlation pattern suggests that the drug Resistome was highly influenced by the growth media.

We first inspected how mutations within the nucleotide synthesis network, the known target pathway, influence drug resistance. *Figure 2A* summarizes the changes observed in barcode representation in this specific gene network. Overall, most gene-knockouts in the core pathways for nucleotide salvage and drug activation increased drug resistance. In contrast, gene-knockouts in the pathway for de-novo nucleotide synthesis increased drug susceptibility. This observation fits well with the known drug toxicity mechanism of antimetabolites that compete with intracellular nucleotides. A closer inspection of this gene network revealed both media and drug dependent features. For example, knockout of either *udp* or *tdk* conferred resistance only in nutrient-rich media (for both drugs) while knockout of *upp* or *uraA* conferred resistance only in nutrient-poor media. Additionally, we observed that strains mutated in the de-novo synthesis pathway were completely missing from all nutrient-poor screens, including the control experiment. This absence reflects the essential role of de-novo nucleotide synthesis if nucleotides are not supplemented in the media. In nutrient-rich media, strains mutated in the de-novo synthesis pathway were detected but were under-represented relative to the control experiment. This observation indicates that knockout of the de-novo nucleotide synthesis pathway increases bacterial drug sensitivity in nutrient-rich conditions.

We next aimed to identify all gene knockouts that confer drug resistance by applying a similar approach to the one that is typically used to identify differentially expressed genes. However, in our implementation we used barcode counts instead of the gene transcripts counts. Specifically, we used the DESeq2 tool (*Love et al., 2014*) that was adjusted for pooled barcode libraries with the DEBRA package (*Akimov et al., 2020*). The full list of barcode frequency and enrichment across screens is presented in *Supplementary file 1*. *Figure 2B* shows how many knockout strains were detected as under-represented (drug sensitive) or over-represented (drug resistant) in each screen

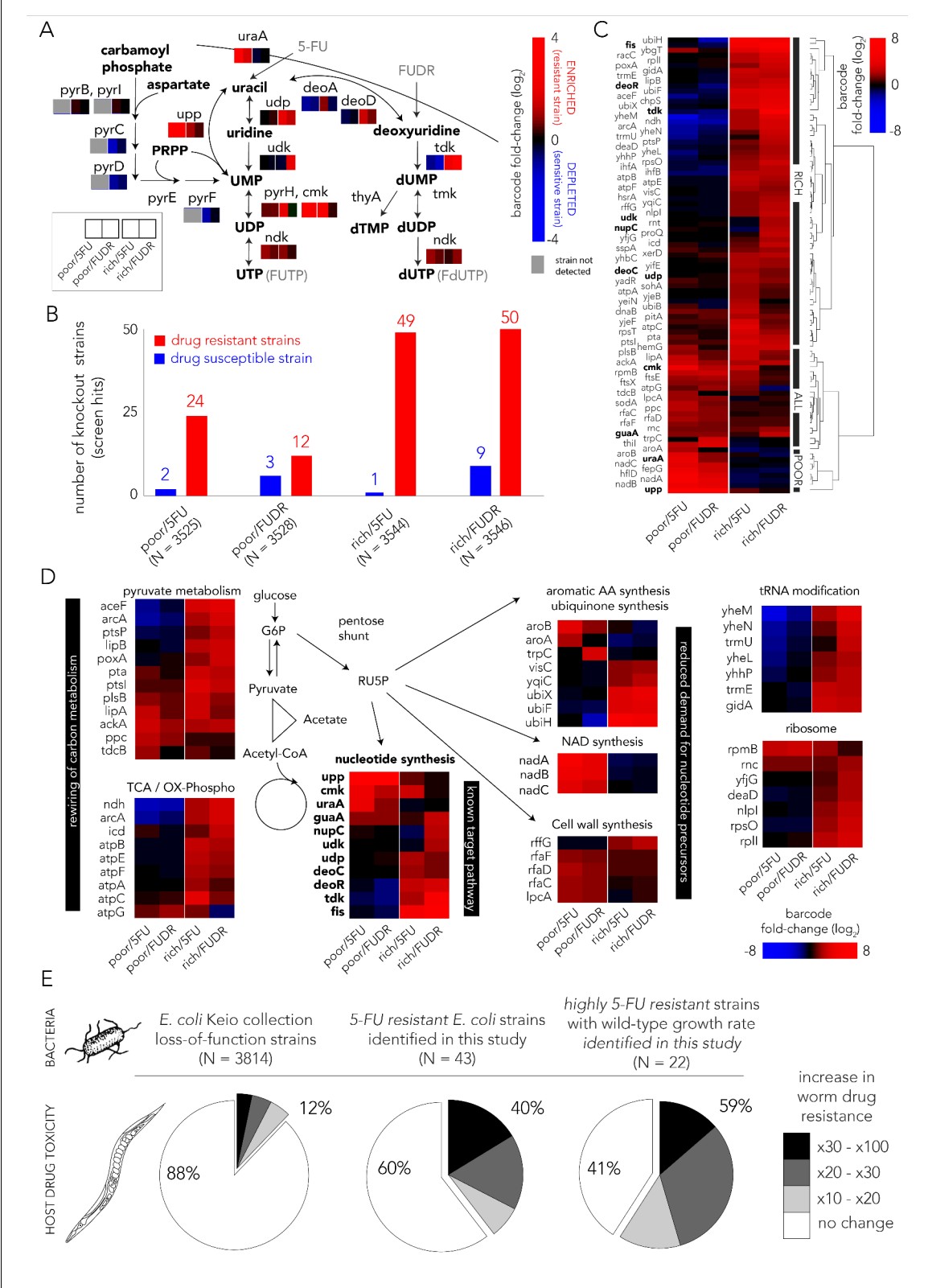

**Figure 2.** Multiple gene knockouts increase drug resistance in bacteria and are expected to lower drug toxicity in the *Caenorhabditis elegans* host. (**A**) Screen results projected on the network of nucleotide synthesis. Compatible with the known toxicity mechanism, we observed that mutations in the salvage and pro-drug activation pathways increase drug resistance (red) while loss-of-function mutations in the de-novo synthesis pathway increase drug sensitivity (blue). (**B**) The number of strains found to be differentially represented in our screens. Hits were determined by fold-change and false-

*Figure 2 continued on next page*

Figure 2 continued

discovery-rate adjusted p-value that was calculated with DEBRA and DESeq2 tools. The number of hits in nutrient-rich media is considerably higher than the number of hits in nutrient-poor media. (C) Unsupervised hierarchical clustering of hits identified in all screen conditions. The clustering uncovers a media-dependent Resistome pattern. The majority of hits are nutrient-rich media specific. Only a minority of hits are nutrient-poor media specific. The genes from nucleotide synthesis network are marked bold. (D) The functional characterization of identified hits reveals a plausible metabolic connection through RU5P, a metabolite from the pentose phosphate pathway that is a precursor for nucleotide synthesis. Hits were assigned to broad functional categories according to their annotated function in the KEGG and EcoCyc databases. (E) Gene-knockouts that increase bacterial resistance will likely reduce drug toxicity in the *C. elegans* host. The three pie charts show the proportion of bacterial strains that reduce drug toxicity in *C. elegans* for different subsets of bacterial strains: all tested strains from the Keio library collection (left) previously tested in the *C. elegans* screen (*Scott et al., 2017*), all 5-FU resistant bacterial strains we identified in our screens and that are present in the *C. elegans* screen (middle), and a subset of highly 5-FU resistant strains we identified (fold enrichment > 6) that also have an un-impaired growth rate in either nutrient-rich or nutrient-poor media (right).

condition (*Figure 1—figure supplement 2* shows the full volcano plots). We estimated that the precision of the genetic screen was around 80% by testing an individual subset of hit strains in 69 growth experiments (*Figure 1—figure supplement 3*). Two interesting trends emerged from this analysis. First, many more knockout strains were identified as resistant strains (12–50) than as sensitive ones (1–9), likely due to the strong negative selective pressure our screens applied. Second, in both drugs considerably more knockout strains emerged as resistant in nutrient-rich media (49–50) relative to nutrient-poor media (12–24). To investigate the mechanisms underlying drug resistance we performed hierarchical clustering on fold-changes in knockout strain representation of the 89 strains identified as hits in at least one condition (*Figure 2C*). This analysis revealed a strong media-dependent structure. The largest cluster grouped 61 knockout strains that were exclusively drug resistant in nutrient-rich conditions. The second largest cluster grouped 20 knockout strains that were drug resistant independent of media type. Lastly, the third cluster grouped eight knockout strains that were resistant exclusively in nutrient-poor media conditions. Interestingly, only 11 genes out of the 89 resistance hits are affiliated with the known target pathway (the gene network of nucleotide synthesis). This observation indicated that additional routes toward resistance exist.

In order to investigate which cellular pathways underlie drug resistance, we used gene set enrichment analysis of the screen results using the GAGE package (*Luo et al., 2009*). We tested for pathway enrichment in GO (*Ashburner et al., 2000*; *Carbon et al., 2018*) and KEGG (*Kanehisa and Goto, 2000*) databases. Beyond the expected enrichment of the nucleotide synthesis network, the analysis revealed enrichment in central carbon metabolism, synthesis of aromatic amino-acids, lipopolysaccharide metabolism, quinone biosynthesis, and tRNA modification. The full list of enriched categories is presented in *Supplementary file 2* and the network of enriched and related GO categories is shown in *Figure 1—figure supplement 4*. Inspection of the annotated functions of the 89 hit genes we previously identified with DESeq2 revealed that 60 of them can be assigned to just to eight enriched pathways (*Figure 2D*). Interestingly, six of these pathways are metabolically linked through the pentose phosphate pathway which synthesizes ribulose-5-phosphate (RU5P), a precursor for nucleotide synthesis. Taken together, these results indicate that resistance can emerge from mutations in multiple pathways, beyond the target network of nucleotide synthesis. Metabolites that are shared between these pathways are one plausible explanation for this effect.

## Potential effect of bacterial 5-FU resistance on the *C. elegans* host

The effect of bacterial gene deletions on 5-FU efficacy in *C. elegans* was tested in two recent studies (*García-González et al., 2017*; *Scott et al., 2017*). We rationalized that a comparison between the *C. elegans* screens and our bacterial screens would allow us to infer if bacterial resistance that will naturally evolve will likely impact the host. It is important to note that this inference is limited to loss-of-function mutations. Specifically, we tested if gene knockouts that are resistant to 5-FU are more likely to reduce drug toxicity in *C. elegans* as measured previously (*Scott et al., 2017*) for another gene-knockout strain collection (*Baba et al., 2006*). Forty-three out of the 69 strains we identified as 5-FU resistant were tested previously as *C. elegans* diet. *Figure 2E* shows the remarkably high overlap that emerges from this comparison: the *C. elegans* screen revealed that only 12% of 3814 tested bacterial strains considerably increased 5-FU resistance in the host (>10-fold); however, this proportion increased to 40% in the subset of 43 hits that confer 5-FU bacterial resistance. Moreover, when

the gene knockouts were restricted only to those that substantially increase bacterial resistance and that do not hamper bacterial growth, this proportion increased to 59% (*Supplementary file 3*). This smaller subset likely better represents the mutations attainable during natural selection since it excludes deleterious mutations that slow growth and it is restricted only to highly advantageous gene-knockouts that tend to fix early in evolving asexual populations (*Wiser et al., 2013*). The observation of such a high overlap is noteworthy since it indicates a fundamental connection between bacterial drug adaptation and host effects – if bacterial evolved resistance will proceed through gene inactivation, it will, in the majority of cases, also reduce drug efficacy in the *C. elegans* host.

## Evolved mechanisms of drug resistance in *E. coli*

The genetic screen uncovered that loss-of-function across multiple pathways can increase drug resistance. However, a screening approach is insufficient in informing which gene inactivation, if any, will transpire when wild-type bacteria evolve under drug selection. Evolution under natural selection may involve more than one resistance mechanism and may leverage on processes beyond loss-of-function mutations such as gain-of-function, over-expression, and neomorphic alleles. The evolutionary adaptation of microorganisms to antimicrobial drugs can be investigated by growing culture in sub-inhibitory drug concentration and serially transferring diluted cultures over multiple days (*Dragosits and Mattanovich, 2013*). While such experiments are conducted in vitro, they can shed light on the molecular mechanisms underlying resistance and the time scale for adaptation. We used a serial transfer protocol to evolve multiple replicates of a wild-type *E. coli* lab strain in inhibitory concentrations of the two drugs in both nutrient-poor and nutrient-rich media (*Figure 3A*). We evolved four independent wild-type *E. coli* strains in each condition over 20 days. We then measured the drug dose–response curves of 10 clones from each independently evolved population in their respective media and drug combination. It is important to note that clones isolated from a single population are not necessarily independently evolving clones and therefore likely share some mutations. To restrict analysis to truly independently evolved adaptations, we continued working only with a single pure clone from each independently evolved population (clones from a single population were typically similar in their drug resistance). We first evaluated the drug sensitivity of evolved strains by testing their growth on a range of drug concentrations and calculating the dose–response curves (*Figure 3B*). As the figure shows, all evolved strains had increased drug resistance. Increases in $IC_{50}$ ranged from about twofold, for adaptation to 5-FU in nutrient-rich media, to more than 60-fold for FUDR adaptation in nutrient-poor media.

The genomic screen we performed with the library of knockout strains uncovered a strong media-dependent pattern of resistance. We therefore decided to test if strains evolved in one condition became resistant to other drug and media combinations. For this analysis we measured the dose–response curve of all evolved clones across all conditions (*Supplementary file 4*). In order to calculate the relative resistance of each clone, we compared their $IC_{50}$ with that of the ancestor (wild-type) strain and of the most resistant strain that evolved in the test condition. This calculation yielded a score that range from zero, marking drug sensitivity comparable to the ancestor, to one, marking drug resistance comparable to the most resistant strain. A summary of this analysis is shown as a heat map in *Figure 3C*. While the analysis revealed multiple instances of cross-resistance in evolved strains, it again uncovered a media-dependent pattern. Strains evolved in a specific media and one drug become resistant to both drugs in that media. In most cases, the strains were not resistant to the same drug they evolved on if it was combined with the other media type (strains that evolved in nutrient-poor media with FUDR where the only exception). This media dependence suggested that the evolved resistance mechanism was likely dictated by the media type.

## Impact of evolved bacteria on the *C. elegans* host

A cross comparison between our 5-FU Resistome and published datasets suggested that the majority of bacterial adaptations through gene inactivation will also reduce 5-FU toxicity in *C. elegans* (*Figure 2E*). We decided to directly test this bacterial-host effect in the 5-FU- and FUDR-evolved bacterial strains. In these experiments we used a bacteria-worm co-culturing method similar to that previously used (*García-González et al., 2017*; *Scott et al., 2017*). *Figure 4A* briefly outlines the approach: evolved bacteria were incubated with L1 animals and media was supplemented with

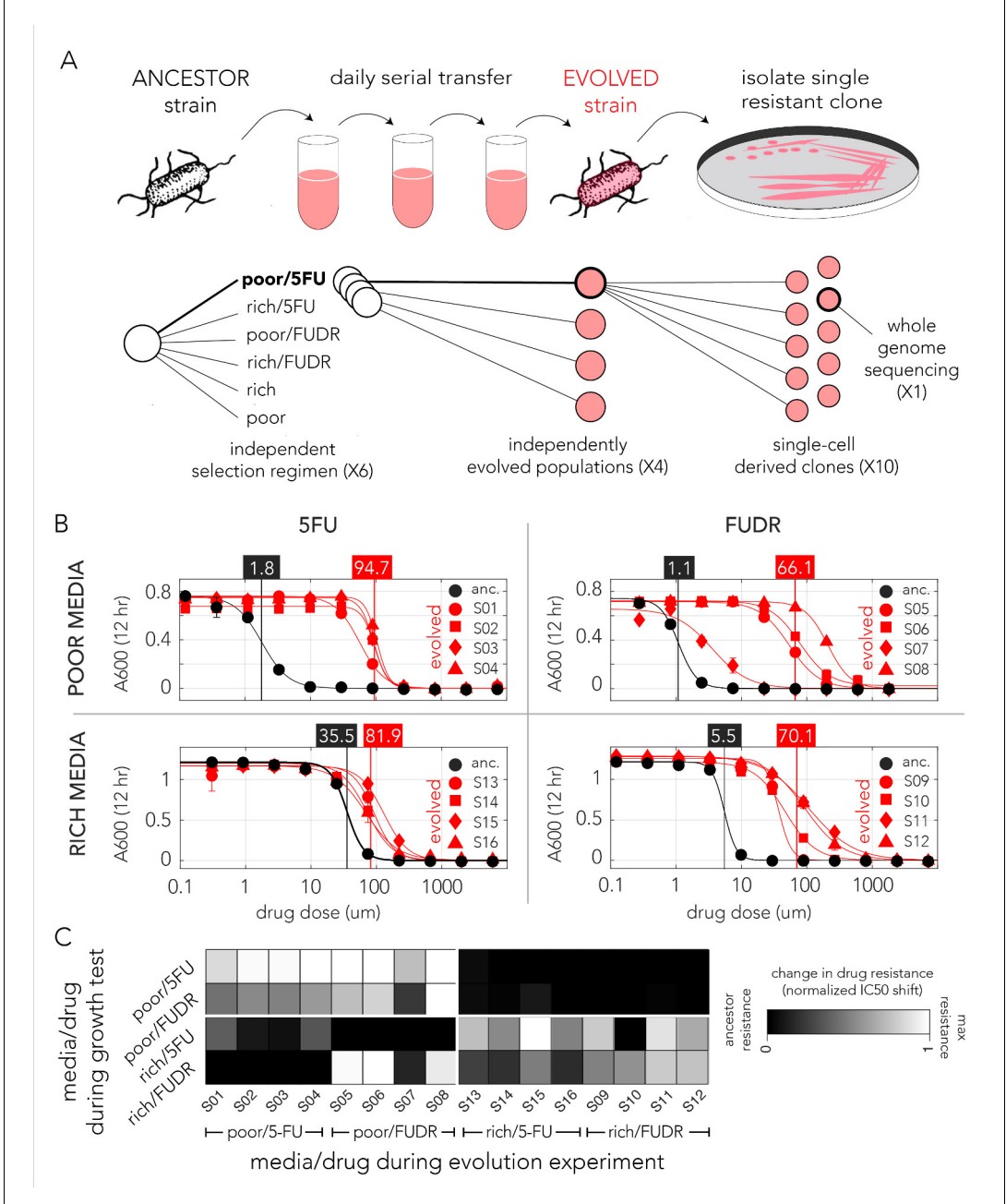

**Figure 3.** Evolved bacteria are drug resistant and display a media-type dependent cross-resistance pattern. (**A**) Overall approach for the lab evolution experiment. We evolved 16 resistant bacterial populations with serial transfer evolution protocol. We evolved four individual replicates on each of the drug and media combinations. We isolated single colonies from each independently evolved population and tested for drug resistance. The lower panel shows the number of replicates used for each step. (**B**) Drug resistance increased in all evolved strains. The curves show the inferred drug sensitivity of clones from four independently evolving populations (red) in comparison to the ancestral strain (black). Increased resistance is reflected by the shift in the $IC_{50}$ to a higher concentration in evolved clones. (**C**) Drug cross-resistance in evolved strains reveals a media-dependent pattern. The heatmap shows the relative change in $IC_{50}$, normalized to the range of $IC_{50}$ in other strains (from the ancestor strain $IC_{50}$ to the maximum measured $IC_{50}$ in strains that evolved on the tested condition).

drugs at varying concentrations. After 60 hr of incubation, the period required for *C. elegans* to reach adulthood, we inspected and scored the developmental stage of the animals (representative microscopy images are shown in *Figure 4B*).

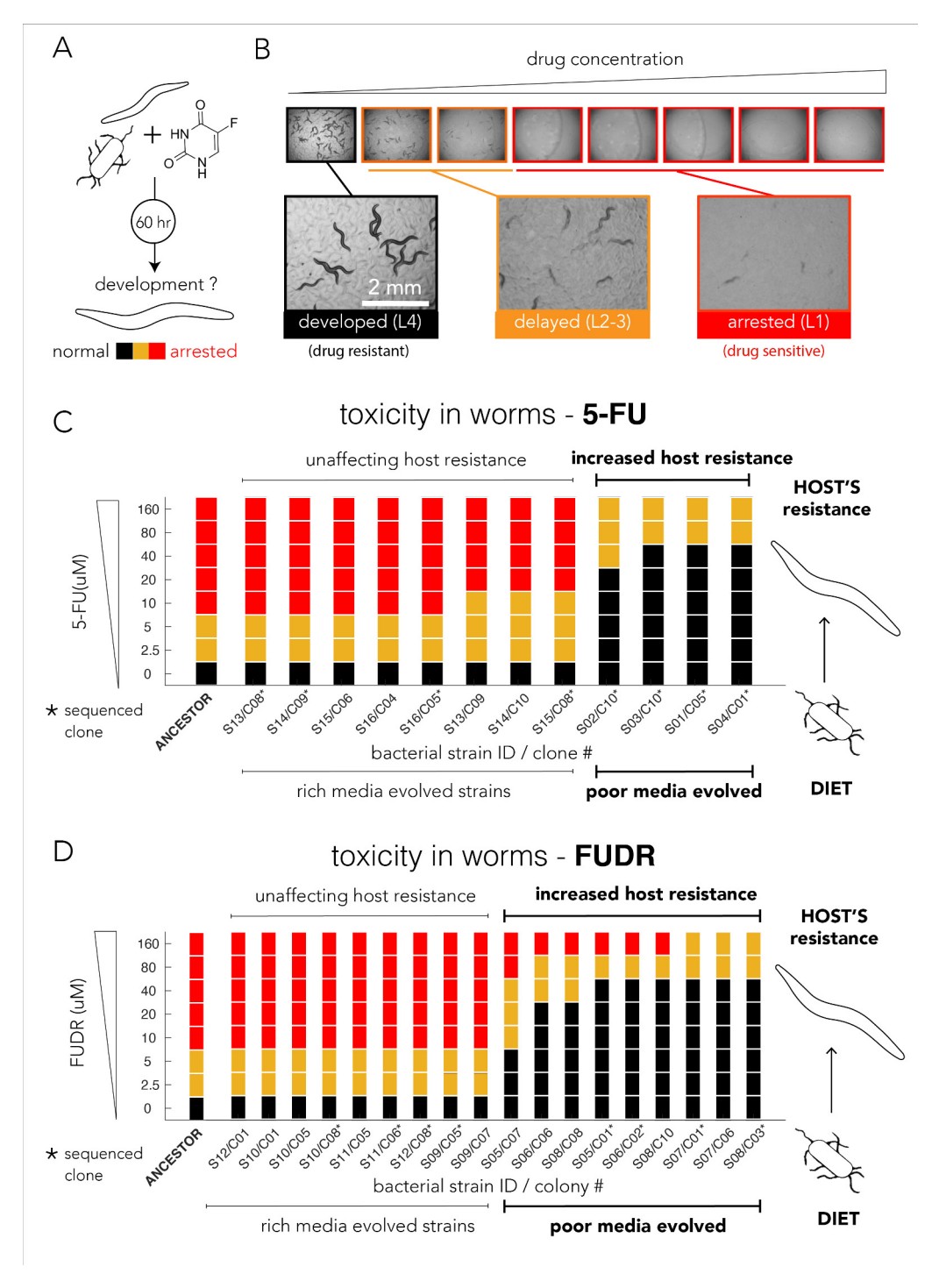

**Figure 4.** Evolved bacterial resistance lowers drug efficacy in the *Caenorhabditis elegans* host. (**A**) Experimental approach for testing the impact of bacteria on drug toxicity in *C. elegans*. The tested bacterial strain was incubated with L1 animals and drug was added in different concentrations. Plates were imaged after 48–72 hr and development stage of the animals was scored into three categories: normally developed (L4 stage), developmentally delayed (L2-3 stages), or completely arrested (L1 stage). Phenotype scoring was performed blindly. (**B**) Representative images showing the development of *C. elegans* growing on different FUDR concentrations when the ancestor strain was used as the bacterial diet. (**C**) Bacteria that evolved on nutrient-poor media and 5-FU reduced 5-FU toxicity *C. elegans*. The color bars indicate the *C. elegans* developmental phenotypes at different 5-FU concentrations and bacterial strains are ordered by their impact on *C. elegans* development. A bacterial strain was characterized as reducing drug efficacy if *C. elegans* development was not completely arrested at the same 5-FU concentration as the animals fed with ancestor strain (20 µM). The set

*Figure 4 continued on next page*

*Figure 4 continued*

of strains that impact the drug efficacy in the host perfectly coincides with their evolutionary history (only nutrient-poor media evolved strains impact host drug efficacy). (D) Bacteria that evolved on nutrient-poor media and FUDR reduced FUDR toxicity *C. elegans*. Similarly, to 5-FU (shown in C), only bacterial strains that evolved in poor-nutrient media impact the host.

We tested drug toxicity in *C. elegans* fed with each of the 30 of the evolved bacterial clones (12 clones evolved on 5-FU and 18 clones evolved on FUDR). These clones were isolated from the 16 individually evolved populations (some of the clones were isolated from the same population). Each clone was tested in three biological replicates and two technical replicates and drug sensitivity results were compared to the drug toxicity in animals feeding on the ancestor strain (*Supplementary file 8*). *Figure 4C* summarizes the results of our experiments for bacterial strains evolved on 5-FU. As the figure shows, 4 of the 12 tested evolved clones considerably lowered 5-FU toxicity in *C. elegans*. Surprisingly, this pattern perfectly coincides with the nutritional history of our evolved clones: only bacterial clones that evolved in nutrient-poor media had an impact on the host (four of four independently evolved clones). Although bacterial clones evolved on 5-FU in nutrient-rich media also became drug resistant, their resistance mechanism had no impact on drug-toxicity in *C. elegans* (eight of eight tested clones), including four of four independently evolved clones. We performed a similar experiment using FUDR and observed an identical pattern. *Figure 4D* summarizes the results of 18 clones from eight independently evolved populations. As the figure shows, 9 of the 18 evolved clones considerably lowered FUDR toxicity in *C. elegans*. Again, this pattern perfectly coincides with nutritional history of evolved strains: only bacterial clones that evolved in nutrient-poor media had an impact on host (nine of nine tested clones) while clones evolving on FUDR in nutrient-rich media had no impact on the host (nine of nine tested clones), including four of four independently evolved clones. In order to test if this perfect match between nutrient availability and host effect is statistically significant given the 16 independently evolved clones from both 5-FU and FUDR conditions, we used a two-tailed Fisher exact test (using evolution history and host effect as categories). The test rejected the hypothesis that nutrient availability during evolution is randomly associated with the host effect (p=0.00016).

Taken together, the *C. elegans* drug sensitivity experiments allowed us to make two important observations. First, the experiment confirmed that under natural selection bacteria frequently evolve and adopt resistance mechanisms that impact host drug sensitivity. This observation concurs with the predictions we made after comparing our genetic screen for bacterial resistance with a published screen that tested *C. elegans* drug sensitivity as a function of bacterial diet (*Figure 2E*; *Scott et al., 2017*). In addition, we observed a striking association between the media used during the evolution experiment and the ultimate effect of the evolved bacteria on the *C. elegans* host. This result highlights the critical importance of the environmental context during natural selection for drug resistance.

## The mutation profile of evolved strains

The phenotypic observations we made in *C. elegans* showed that only some of the evolved bacteria strains reduced drug toxicity in the host and therefore strongly suggested that bacterial evolution took advantage of at least two alternative drug resistance mechanisms. We therefore decided to sequence the genomes of evolved clones in order to identify all mutations that emerged during this short evolution experiment. *Figure 5A* summarizes the type of mutations we observed according to individual mutations. Across 16 sequenced genomes we identified 75 mutations affecting 49 genes, 80% of them within coding sequences. Based on the predicted effect of the mutation on the nearby open reading frame we classified mutations as ones that cause gene loss-of-function or ones that potentially only modify the gene by altering its coding sequence or promoter (*Figure 5A*). We predicted that at least 55% of all mutations lead to gene loss-of-function (large deletions, transposon insertions, and premature stop codons). This estimation is a lower bound of the total proportion of gene inactivation mutations since additional mutations such as point mutations and promoter deletions can also confer loss-of-function.

Mutations are expected to be distributed randomly across the genome and natural selection is expected to fix those that are advantageous to the organism under the relevant growth condition. However, other forces, such as genetic hitchhiking and genetic drift, are expected to lead to fixation

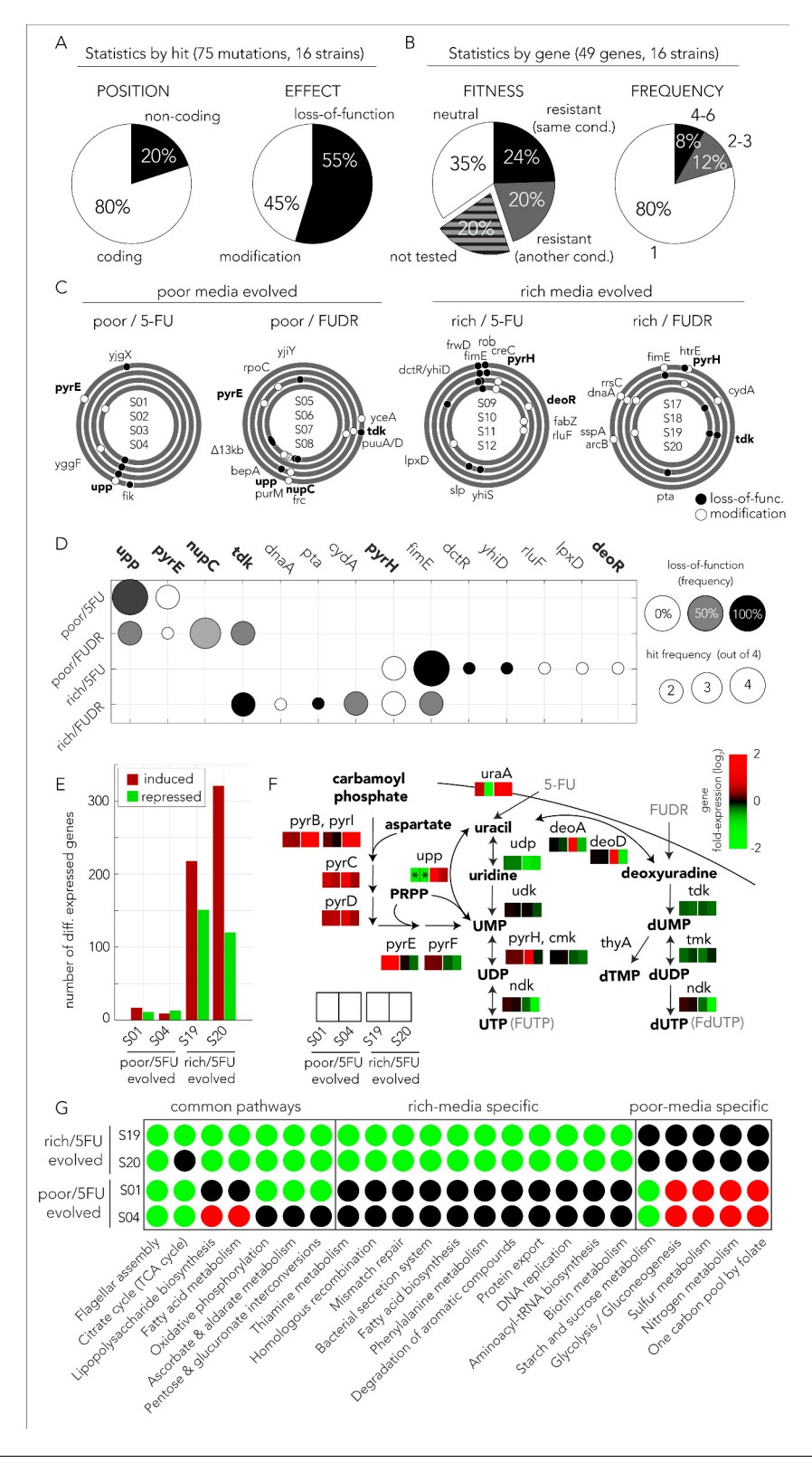

**Figure 5.** The evolved mechanisms underlying bacterial drug resistance are media-type dependent. (**A**) Characterization of mutation positions and effects on the nearest open reading frame. Mutations that clearly disrupt the reading frame are annotated as loss-of-function mutations (frameshift mutation, large deletion, or transposon insertion). Other mutations are annotated as modifying mutations (point mutations, in-frame indels, and promoter mutations). (**B**) Characterization of mutations by their effect on genes. Mutations are annotated as leading to resistance if a gene-knockout

*Figure 5 continued on next page*

*Figure 5 continued*

strain is drug resistant. (C) The position of all mutations in the genomes of the sequenced evolved clones. The concentric circa plots show the genomes of four clones taken from four individually evolved populations. Circles represent the annotated strains from inner to outer order. Black circles mark loss-of-function mutations and white circles mark gene modifying mutations (similar to A). The genes from nucleotide synthesis network are marked bold. (D) Summary table of putative driver mutations across all sequenced strains. A mutation was characterized as a driver mutation if a loss-of-function of the corresponding gene was observed to significantly increase or decrease drug resistance. (E) Number of differentially expressed genes in four 5-FU resistant strains growing without any drug. The nutrient-rich media evolved strains showed a significantly higher number of differentially expressed genes relative to the strains that evolved on nutrient-poor media. (F) Changes in gene expression in evolved strains projected on the network of nucleotide synthesis. Asterisks signs mark loss-of-function mutations detected during whole genome sequencing. (G) Differentially expressed KEGG pathways in 5-FU-evolved strains. The circles mark individual pathways that were identified as significantly induced (red), or repressed (green), by the GAGE tool.

of neutral mutations as well. We therefore expected that only a subset of observed mutations would confer drug resistance. We used two methods to identify which mutations were likely advantageous using a gene-focused approach (*Figure 5B*). We first tested if a gene knockout strain, corresponding to a gene in which mutations were identified in the evolution experiment, was more drug resistant than the wild-type strain. A total of 39 loss-of-function strains from the Keio deletion library overlapped with the 49 genes that were mutated in evolved strains. Roughly one-third of the knockout strains tested (12 of 39) showed increased drug resistance (left pie chart in *Figure 5B*). We therefore inferred that mutations in 27 genes are either hitchhiker mutations or mutations that confer an advantage through gain-of-function, rather than a loss-of-function. As a complementary approach for identifying advantageous mutations, we tested if individual genes were repeatedly hit in independently evolved populations (right pie chart in *Figure 5B*). We observed that 10 of the 49 genes were hit at least twice and four of those were hit more than four times across the 16 sequenced clones. A full summary of the mutations and validated loss-of-function resistance are detailed in *Supplementary file 5*. Although the two approaches for detecting advantageous mutations were different, they identified a highly overlapping set of eight common genes. In order to test if this high overlap is statistically significant, we used a one-tailed Fisher exact test while assigning genes to the categories adaptive/non-adaptive according to each one of the methods. The test rejected the hypothesis that the two methods are randomly associated (p=0.0017). Taken together, we predict that a total of 14 genes out of the 49 observed mutated genes were selected for in our evolution experiment.

## Evolved mutations influence multiple resistance mechanisms

We observed that 49 genes were mutated across all conditions. Next we tested if mutations in specific genes depended on the media and drug type during the evolution experiment. *Figure 5C* shows the genomic location and type of all 75 identified mutations across all sequenced genomes. By displaying the genome of evolved replicates as concentric circles we could easily identify genes that were repeatedly mutated. For example, the nucleoside transporter NupC was mutated in three replicates that evolved in FUDR and nutrient-poor media. The observation that *nupC* was exclusively mutated in that condition suggested it is involved in FUDR transport. This makes sense, because NupC is a known nucleoside transporter, and because FUDR is a nucleoside analog. To better identify drug-specific and media-specific mechanisms we grouped hit genes according to their evolutionary history (*Figure 5D*). We observed that mutated genes were grouped predominantly by media type (four of five). Specifically, in nutrient-poor media we observed that *upp*, a gene central for nucleotide salvage, was frequently inactivated (six of eight). Additionally, we observed that *pyrE*, an essential gene for pyrimidine de-novo synthesis, was frequently modified (three of eight). In contrast, evolved resistance in nutrient-rich media frequently (four of eight) selected for modification of the *pyrH* gene, a UMP kinase involved in drug activation and a regulator of de-novo nucleotide synthesis (*Kholti et al., 1998*). In nutrient-rich conditions, we also observed frequent (six of eight) inactivation of *fimE*, a regulator of *fimA* involved in cell adhesion. Only one gene was characterized by a drug-specific pattern – *tdk* coding for kinase of deoxythymidine and deoxyuridine was frequently (four of eight) inactivated in FUDR-, but not 5-FU-, evolved strains.

The set of mutated genes (*Figure 5D*) showed that evolved resistance leveraged on multiple mechanisms of resistance. Mutations influencing the network of nucleotide synthesis were found

across all conditions suggesting that resistance emerged from reduced pro-drug activation by hampering the nucleotide salvage pathway (*upp*, *nupC*) and drug metabolism (*tdk*, *deoR*). In addition, resistance was also likely to improve due to a compensatory increase in de-novo nucleotide synthesis (modifying mutations in *pyrE* and *pyrH*). Interestingly, mutations in additional genes suggest that other mechanisms were also involved, including central carbon metabolism (*pta*, *cydA*, and *dctR*), rRNA modification (*rluF*), and cell-wall synthesis (*lpxD*). Mutations in all these additional pathways agree with the observations we previously made in our genetic screen that showed the loss-of-function of genes in these pathways confer resistance (*Figure 2D*). The mechanisms underlying resistance for two genes, *fimE* and *yhiD*, remain unclear. Previous work revealed that inactivation of the regulator *fimE* leads cells to shift their lifestyle from planktonic to pellicle-like growth in broth cultures without reducing the growth rate (*Matange et al., 2019*; *Stentebjerg-Olesen et al., 2000*). Pellicle formation by *fimE* inactivation was also observed to be selected for in *E. coli* cells exposed to the antibiotic rifamycin and was found to improve resistance by increasing cell aggregation due to increased cell adhesion (*Matange et al., 2019*). It is therefore possible that a similar mechanism also improves 5FU and FUDR resistance and can therefore explain the frequent inactivation of *fimE* in our evolution experiments. Taken together the genome sequencing results revealed that evolutionary adaptation toward drug resistance was highly influenced by the media type and leveraged on multiple mutations in additional pathways beyond the network for nucleotide synthesis.

## Transcriptional changes in 5-FU-evolved strains

The genome sequencing revealed that adapted clones harbor only a handful of putatively driver mutations. However, since many of these mutations modify central metabolic genes and gene regulators, they may culminate in transcriptional changes that extend far beyond the mutated genes. We rationalized that in some cases a few mutations can lead to transcriptional reprogramming of hundreds of genes that, in turn, involves additional drug resistance mechanisms. In order to explore transcriptional reprogramming in response to drug exposure, we compared the transcriptional program of 5-FU-evolved strains to the ancestor strain using mRNA sequencing (RNA-seq). We decided to focus on 5-FU-evolved strains since the mutational patterns underlying drug resistance were mostly media-driven and not drug-driven. In these experiments we grew four of the 5-FU resistant strains, two that evolved on nutrient-poor media and two that evolved on nutrient-rich media, on the same media they evolved on. Since the mutations we observed mostly occur in house-keeping and not in stress-response genes, we decided to inspect changes in the basal transcriptional program and therefore preformed all experiments without any drug.

We first tested how many genes were differentially expressed in evolved strains relative to the ancestor strain using DESeq2 (*Figure 5E*, *Supplementary file 6*). This analysis revealed that an extensive transcriptional reprogramming, involving hundreds of genes, took place in nutrient-rich media evolved strains. In contrast, strains that evolved on nutrient-poor media underwent a modest reprogramming that involved only a few dozens of genes. We first focused on transcriptional changes in genes from the core network for nucleotide synthesis (*Figure 5F*). As the figure shows, we observed that the transcriptional changes in the target gene network agreed with our interpretation of the mutational pattern and aligned with expectations from the genetic screen: all evolved strains overexpressed the pathway for de-novo nucleotide synthesis and overall inhibit the pathway for nucleotide salvage and pro-drug activation, albeit through different genes. Poor media evolved strains reduced nucleotide salvage through *upp* knockout while nutrient-rich media evolved strains repressed *udp* expression. Reassuringly, although *upp* and *udp* are on parallel arms for drug activation, this media dependency matched the media-dependent pattern we saw in the genetic screen for this gene pair (see *upp* and *udp* on *Figure 2A*). Lastly, nutrient-rich media evolved strains also inhibited the expression of two additional kinases (*ndk* and *cmk*) involved in drug phosphorylation.

In order to unbiasedly test for reprogramming in additional pathways we tested for gene set enrichment using the GAGE tool. *Figure 5G* shows the summary of the KEGG pathways with modified expression (a full list of affected pathways appears in *Supplementary file 7*). We reassuringly observed that many of the resistance pathways identified in our genetic screen were repressed in the evolved strains. For example, central carbon metabolism was repressed in all strains (see oxidative phosphorylation and citrate cycle). Additionally, rich media evolved strains repressed multiple additional potentially relevant pathways identified in our screen, such as lipopolysaccharide biosynthesis and phenylalanine metabolism. Interestingly, nutrient-poor media evolved strains induced the

glycolysis/gluconeogenesis pathway, potentially increasing flux into the pentose phosphate pathway. Lastly, the analysis also uncovered changes in additional pathways not associated with drug resistance in the genetic screen. These changes may either reflect adaptation to the growth media or additional mechanisms that increase drug resistance. For example, the repression of flagellar assembly, observed in all evolved strains, may increase drug resistance by inactivation of a highly demanding cellular process that is unneeded in a serial transfer experiment. Taken together, the experiments revealed that broad transcriptional changes took place and involved up to hundreds of genes from multiple pathways. The pattern of transcriptional changes concurs with results from our genetic screens and reveals that many of the pathways that increase resistance upon gene knockdown are overall repressed in evolved strains. The compatibility of these transcriptional changes with our genetic screen suggests that changes in the basal transcriptional program are likely adaptive and will increase resistance with drug encounter. However, we cannot rule out that additional changes in gene regulation, as gene induction or repression that occurs only with drug exposure, also exist.

## Discussion

Many current research efforts are dedicated to correlating microbiota species-composition with their metabolic capacity in order to ultimately predict microbiome impact on drug efficacy. However, such approaches overlook a key characteristic of microbial systems – the ability of individual species to rapidly evolve and change. Here we propose that since many host-targeted drugs impair microbiome growth (*Maier et al., 2018*) and since bacteria can rapidly adapt within hosts (*Barroso-Batista et al., 2014*; *Crook et al., 2019*; *Lourenço et al., 2016*), it is possible that intraspecies evolution will have a significant impact on drug efficacy in the host. We provide a proof-of-concept for this claim in a model system by showing that evolved resistance in *E. coli* against two fluoropyrimidines can select for mutations that reduce drug sensitivity in the *C. elegans* host. As previously proposed, working on a simple model system may be a valuable first tool for disentangling of complex host–drug–microbiome interactions that exist in humans (*Douglas, 2019*; *Maier and Typas, 2017*). Importantly, the conclusions of our work may have implications for evaluating the role of the microbiome in the clinical setting since it demonstrates that information on microbiome species composition alone can be insufficient to evaluate the microbiome's impact on host–drug interactions. Specifically, our work suggests that some chemotherapy treatments can lead to adaptive mutations in individual species of the microbiome that alters drug metabolism. Such bacterial adaptations are independent from changes in microbiome composition and should therefore be monitored as well.

We started by systematically exploring which loss-of-function mutations increase drug resistance with a genetic screen (*Figure 1D*). This approach expanded our understanding of the cellular mechanisms involved in microbial drug resistance. Compatible with the known toxicity mechanisms in *E. coli*, our screen confirmed that knockouts in the pathways for nucleotide salvage, pro-drug activation, and cell-wall synthesis increase resistance (*Tomasz and Borek, 1962*; *Tomasz and Borek, 1960*). However, the screen also pointed to multiple additional pathways that were not previously associated with resistance (*Figure 2D*). Many of these pathways can potentially influence resistance by increasing the supply or reducing the demand for RU5P, a precursor for nucleotide synthesis. Another potential mechanism common to many of the identified pathways is their inhibitory effect on oxidative phosphorylation. Such inhibition was previously observed to increase drug resistance against many antibiotics (*Charbon et al., 2017*; *Lobritz et al., 2015*). However, beyond uncovering putative resistance mechanisms, our screen also revealed that resistance mechanisms were dissimilar in nutrient-poor and nutrient-rich environments. The observations made in both types of media are important given that microbiota are found in multiple sites in the human body that significantly differ in nutrient availability. For example, the gut is a nutrient-rich environment that is likely abundant in nucleotides (*Tramontano et al., 2018*) while other microbiome sites, such as the skin and reproductive organs, may not be so nutrient rich. Bacteria are also frequently found in solid tumors (*Geller et al., 2017*; *Nejman et al., 2020*; *Riquelme et al., 2019*) which themselves harbor a range of microenvironments with varying amount of nutrients due to disrupted tissue organization and blood supply as well as necrotic regions (*Balkwill et al., 2012*; *Carmona-Fontaine et al., 2017*).

Taken together, the screen results allowed us to make important predictions that most likely also hold beyond this model system. First, that resistance against the two tested drugs can emerge

rapidly under natural selection since substantial resistance can already emerge with a single gene inactivation. Second, that nutrient availability will play an important role in determining the bacterial resistance mechanisms that will ultimately evolve. Lastly, a comparison between our screen and a previous screen that tested drug toxicity in *C. elegans* as a function of its bacterial diet uncovered a remarkable overlap in the hit genes (*Figure 2E*). This overlap raises a thought-provoking suggestion that is now also backed by quantitative observations that many of the bacterial drug adaptations (transpiring through gene loss-of-function) are expected to also reduce drug toxicity in the host.

We next tested the predictions from our genetic screen by conducting a short evolution experiment. While this experiment was conducted in vitro, it is still valuable for uncovering the resistance mechanisms that can evolve in short evolutionary time scales in response to drug selection. Indeed, as was expected by the screen results, bacteria rapidly became drug resistant (*Figure 3B*) and they displayed a cross-resistance pattern that clusters by media type (*Figure 3C*). Excitingly, when we tested the effects of evolved strains on drug toxicity in the *C. elegans* host, we detected a similar dependency on nutrient availability during the evolution period – only strains that evolved on nutrient-poor media impacted the host (*Figure 4C and D*). This phenotypic observation was important for two reasons: it supported our hypothesis that evolved drug resistance in bacteria will frequently impact the host and it revealed that the evolved adaptation mechanism was channeled by nutrient availability during the selection period. Genomic sequencing of evolved strains revealed the molecular underpinning for this media-dependent pattern and further corroborated our conclusions from the genetic screen. Here, again, we observed a non-random mutation pattern that clusters primarily by nutrient availability during evolution (*Figure 5D*). Moreover, evolved resistance often involved more than one resistance mechanism in each evolved clone. For example, we observed both the inactivation of nucleotide salvage and induction of de-novo nucleotide synthesis in strains evolved in nutrient-poor media and 5-FU (*Figure 5C*). This conclusion is further supported by genome wide transcriptome analysis of 5-FU-evolved strains that revealed transcriptional reprogramming of multiple pathways beyond the ones that were mutated (*Figure 5G*). Loss of function in many of these pathways was associated with increased resistance in our genomic screen (*Figures 2D* and *5G*).

The mechanistic model presented in *Figure 6* can potentially explain our observations on 5-FU-evolved bacteria and their host effects (we believe a similar model also applies for FUDR). In the ancestor strain, the pro-drug is imported and is converted to various toxic intermediates, with FUMP being especially toxic for the *C. elegans* host (*García-González et al., 2017*; *Scott et al., 2017*; *Figure 6*, upper panel). Bacteria evolving in 5-FU and nutrient-rich media adapt by repressing genes involved in pro-drug salvage and activation (*udp*, *cmk*, and *ndk*). However, strong suppression of the salvage pathway is unfavored since the media is rich in nutrients that are still beneficial for the cell. Since many of the adaptations in nutrient-rich evolved strains are downstream to the FUMP intermediate, drug toxicity is low for bacteria but remains high for the *C. elegans* host (*Figure 6*, middle panel). To counteract reduced nucleotide salvage, evolved strains also repress oxidative phosphorylation and overexpress genes in the de-novo nucleotide synthesis pathway. The latter effect is likely facilitated by the mutation in *pyrH*, both a UMP kinase and a known regulator of the de-novo nucleotide synthesis (*Kholti et al., 1998*). Evolution in nutrient-poor media conditions involves overall similar bacterial adaptation but is facilitated by different genes, and thus it affects the *C. elegans* host differently (*Figure 6*, lower panel). Most importantly, in these evolved strains, nucleotide salvage is practically shut down due to inactivation of *upp*. In a nutrient-poor environment, *upp* knockout is likely to have a marginal cost. In addition, mutations in *pyrE* promoter potentially increase its expression and improve resistance by increasing the flux through the de-novo nucleotide synthesis pathway. This increased flux is further supported by the changes in the central carbon metabolism network. Importantly, since Upp is upstream to the FUMP intermediate, its inactivation alleviates FUMP accumulation and reduces the toxic impact of 5-FU on *C. elegans* (*García-González et al., 2017*).

The microbiome is a key force influencing disease progression and treatment success. While numerous studies have tested for correlations between microbiome species composition and treatment success, they typically overlook mutations and adaptation at the individual species level. Despite some advancement, the high complexity of microbiome interactions with the human host remains a formidable challenge that impedes advancement. Model organisms have proven to be useful tools for resolving the underlying principles that dictate complex host–microbiome interactions (*Douglas, 2019*; *Maier and Typas, 2017*). However, it is also important to acknowledge the

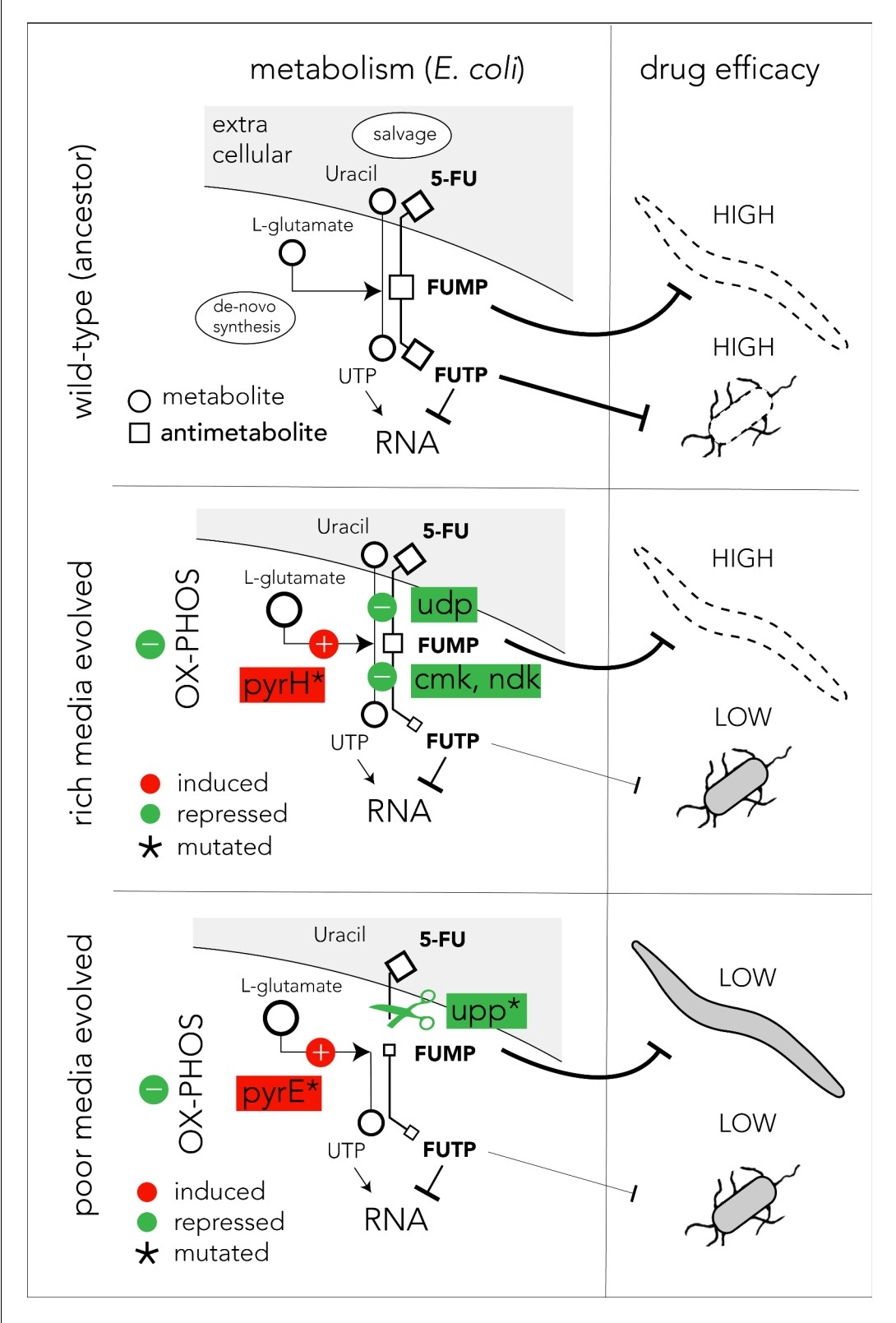

**Figure 6.** Proposed model of evolutionary adaptation to 5-FU. The nutrient availability during the period of 5-FU selection leads to alternative adaptation mechanisms in bacteria and culminate in different outcomes in the *Caenorhabditis elegans* host. The top panel shows the drug toxicity mechanism in *C. elegans* feeding on the ancestor bacterial strain. Although evolutionary adaptation always selected for resistant bacteria, different

*Figure 6 continued*

mechanisms of adaptation lead to different impacts on the host (middle and lower). The genes that were repeatedly mutated in evolved strains are marked with an asterisk sign.

inherent limitations of using simplified model systems and in vitro methods to study the highly complex microbiome–drug–host interactions that may exist in humans, such as the inability to mimic host physiology, to capture species-level and strain-level microbial variability, and to recreate the rich interactions that exist between microbiome members.

Our study investigated the cellular mechanisms underlying bacterial evolutionary adaptation to a chemotherapy drug that is common in clinical oncology and uncovered a surprising phenomenon with potentially important implications. Our work suggests that the microbiome's impact on drug–host interaction will not be static but may dramatically shift if bacteria adapt and become resistant to the host-targeted drug. While our model system is much removed from humans, a similar evolutionary principle may also transpire in the human microbiome, either in its natural sites or within infected tumors. Taken together with the recent realization that numerous host-targeted drugs inadvertently also hinder microbiome growth, bacterial adaptation to host-targeted drugs may not be rare. Our work reveals that since bacterial adaptation may transpire through changes in bacterial drug metabolism, it may have implications for personalizing drug therapy according to the patient's microbiome.

# Materials and methods

## Media and growth conditions

All bacterial experiments in the study were performed in either nutrient-poor media (M9 minimal medium supplemented with 0.2% amicase and 0.4% glucose) or nutrient-rich media (Luria Broth, LB). *C. elegans* experiments were conducted using standard nematode growth medium (NGM) at 25°C. Bacterial cultures of strains with an antibiotic resistance cassette were incubated overnight on selective media. The media for strains from the Keio knockout collection (*Baba et al., 2006*) was supplemented with 50 µg/ml kanamycin and the media for the barcode strain collection was supplemented with 20 µg/ml chloramphenicol. Bacterial experiments were performed at 37°C, and overnight cultures were shaken (orbital) at 200 rpm. Bacterial experiments in multi-well plates were conducted in a plate reader at 37°C with double orbital shaking.

## Bacterial drug-response curves

Bacteria were grown overnight in their respective media conditions (poor/rich) after inoculation from a frozen glycerol stock or single colonies. Overnight cultures were diluted to $OD_{600}$ of 1 before starting the experiments. Cultures were then diluted 1:200 into pre-prepared 96-well plates containing 200 µl of media supplemented with drug. We used the following serial drug dilutions: twofold dilution of 5-FU/rich media starting at 18 mM and FUDR/poor media starting at 1.8 mM, and threefold dilution of 5-FU/poor and FUDR/rich media starting at 7.3 mM each. We monitored changes in $A_{600}$ (absorbance, 600 nm) with an automated plate reader (Tecan Spark/BioTek Eon MicroPlate) at 10 min intervals for 12 hr. We used the background subtracted absorbance ($A_{600}$) values after 12 hr of growth to calculate $IC_{50}$ in each experiment using a custom MatLab code (MathWorks). All experiments were done as biological duplicates.

## Drug resistance of individual knockout strains

Strains were grown overnight in their respective media conditions (poor/rich) after inoculation from a frozen glycerol stock or single colonies. Overnight cultures were diluted to $OD_{600}$ of 1 before starting the experiments. In a 384-well plate, cultures were diluted 1:200 in 75 µl of respective drug/media for validation. We monitored changes in absorbance ($A_{600}$) with an automated plate reader (Tecan Spark/BioTek Eon MicroPlate) at 20 min intervals for 12 hr.

## Growth rate calculation and classification of slow growing strains

Bacteria were grown overnight in their respective media conditions (poor/rich) after inoculation from a frozen glycerol stock or single colonies. Overnight cultures were diluted to $OD_{600}$ of 1 before starting experiments. Cultures were diluted 1:200 and 200 µl aliquots were transferred into each well in a 96-well plate. We monitored changes in absorbance ($A_{600}$) with an automated plate reader (Tecan Spark/BioTek Eon MicroPlate) at 10 min intervals for 12 hr. We used the background subtracted absorbance values in the logarithmic growth phase ($0.01 < A_{600} < 0.15$) to fit an exponential growth curve and calculated the generation time using a custom code in MatLab (MathWorks). We used this protocol to quantify the growth rates of the ancestor strains and all evolved strains in nutrient-rich and nutrient poor media. All experiments were done as biological duplicates and the mean growth rate was used. We used a similar approach to classify individual knockout strains as slow growing strains from the Keio strain collection. In these experiments we grew the culture in a 384-well plate (75 µl aliquots of culture) to allow simultaneous measurements of biological replicates from almost a hundred different strains. Knockout strains with a generation time that was higher by 25% relative to the wild-type strain were classified as slow growing strains.

## Drug toxicity in *C. elegans*

The drug toxicity in *C. elegans* was quantified by characterizing the development stage of animals fed with different *E. coli* strains in the presence of 5-FU or FUDR. For both drugs we tested the 2.5–160 µM concentration range with a twofold serial dilution. All screens were performed in a 48-well NGM agar plates with three biological replicates (individual wells) and two technical replicates (imaged areas in the well). We used the N2 (Bristol) *C. elegans* strain. All 30 evolved *E. coli* clones and the ancestor strain were inoculated directly from frozen glycerol stocks into 800 µl LB for overnight growth. Cultures were concentrated 2× and 50 µl was added to agar covered wells containing NGM and drug. Plates were left at room temperature overnight. Worms were synchronized by bleaching gravid animals to harvest the eggs. Approximately 20–40 synchronized L1 animals were added to NGM plus drug and *E. coli* wells and incubated at 20°C for 48–72 hr until the no-drug animals reached the L4 stage. Animal development phenotypes were visually identified by microscopy at 2× magnification. The experimenter classifying the developmental phenotype was kept blind to the identity of the bacterial strain used in each well.

## Barcoded strain library

The *E. coli* barcoded deletion library was developed and provided by HM. Full details of this resource will be published elsewhere. The parent strain of this library is BW38028 with the genotype Δ(araD-araB)567 lacZp-4105(UV5)-lacY λ⁻ hsdR514, rph+ (*Conway et al., 2014*). The strain collection has 3680 individual knockout strains. In each strain the open reading frame of a single gene was replaced in-frame with a fragment containing turbo GFP, chloramphenicol resistance cassette, and a unique 20 bp sequence that serves as an identification barcode. Since the barcode is the only variable region across strains, it can be amplified from a mixed culture of strains with a single pair of primers. We used primers that amplify a 325 bp region.

## Pooled genetic screen

We first performed a drug sensitivity experiment using *E. coli* BW25113 and calculated the $IC_{90}$ for 5-FU and FUDR in nutrient-poor and nutrient-rich media. For nutrient-rich media, we used 100 µM 5-FU and 24 µM FUDR, and for nutrient-poor media we used 2.7 µM 5-FU and 2.3 µM FUDR. We inoculated 15 µl of the glycerol stock of the pooled strain library into 25 ml of the two media types for overnight growth (this high inoculum was used to maintain a high population size and avoid a sampling bottleneck in the thawing stage). In the morning we diluted the culture to $OD_{600}$ of 1 and again diluted them 1:200 into 7 ml media supplemented with drugs to start the screen. We performed the screens in biological duplicates. We monitored the optical density of cultures throughout the screen. When the cultures reached $OD_{600}$ of 0.6 we stopped the experiment and immediately extracted DNA (Zymo Quick DNA Miniprep Plus Kit, Cat#D4068).

## Barcode sequencing and analysis

We measured the genomic DNA (gDNA) concentration of each screen sample with Qubit dsDNA high sensitivity assay kit (Thermo-fisher, Cat#Q32854) and used 6.25 ng of gDNA to prepare the DNA libraries. A region of ~350 bp around the barcode locus was amplified with custom forward and reverse primers using 2× KAPA HiFi HotStart ReadyMix (Kapa Biosystems, Cat#KK2602). The primer sequences were:

> 5' TCGTCGGCAGCGTCAGATGTGTATAAGAGACAG-(4-6xN)-TGTAGGCTGGAGCTGCTTCG 3'
> 5' GTCTCGTGGGCTCGGAGATGTGTATAAGAGACAG-GCAAATATTATACGCAAGGCGA-CAAG 3'

The following thermocycler protocol was used: initial denaturation at 95°C for 3 min, 23 cycles of 95°C for 30 s, 55°C for 30 s, 72°C for 30 s, followed by a final extension at 72°C for 5 min. PCR products were purified using standard AMPure XP bead protocol (Beckman Coulter, Cat#A63881) with beads added at 0.9× volume. The standard Nextera XT Index Kit protocol (Illumina, Cat#FC-131–1024) was used to add indices and Illumina sequencing adapters to each PCR sample followed by the same AMPure XP bead purification protocol. The libraries were then run on a 2.5% agarose gel and the product was extracted using NEB Monarch DNA Gel Extraction Kit standard protocol (NEB, Cat# T1020L). Quality control of libraries was performed using BioAnalyzer/Agilent High Sensitivity DNA Kit (Agilent Technologies, Cat# 5067–4626). Library concentrations were assessed by Qubit dsDNA high sensitivity assay. Libraries were normalized to be in the same concentration, denatured, and diluted according to Illumina MiniSeq System Denature and Dilute Libraries Guide. Sequencing was performed using MiniSeq High Output Reagent Kit, 75-cycles (Illumina, Cat# FC-420–1001) on Illumina MiniSeq device. Raw reads were converted to barcode counts using a custom MatLab (MathWorks) script that searched for exact 15–25 bp barcode matches in each individual read. We masked any nucleotide with a quality score of 10 or less. We identified barcodes in 85% of all reads. Analysis of the remaining 15% of the reads (that did not have a matched barcode) showed that 60% them include barcode homologs (sequences that can be matched to a barcode with a single mutation). Since identical barcode homologs were found across all screen conditions we inferred that they existed in bacteria prior to the screen and likely originated from library construction (from errors in primers, amplification, or cloning). We do not expect that these unassigned reads are impacting the screen results.

We identified gene knockouts that influenced bacterial drug sensitivity by comparing the relative frequency of individual barcodes when the pooled library grew in the presence of the drug and in a no-drug control experiment. For this analysis, we used the barcode counts and identified barcodes with significant changes in their relative frequency with DEBRA (*Akimov et al., 2020*). DEBRA is a package that allows the running of DESeq2 (*Love et al., 2014*) analysis on barcode libraries. For this analysis we chose the Wald statistical test and cutoffs of fourfold for enrichment and false-discovery-rate adjusted p-value of 0.1. We discarded barcodes with less than 10 counts. We performed Gene-set enrichment analysis with GAGE (*Luo et al., 2009*) with a false-discovery-rate adjusted p-value of 0.1.

## In vitro adaptive evolution experiment

We used the serial transfer method to evolve bacterial drug resistance in vitro (*Dragosits and Mattanovich, 2013*). For these experiments we used the BW25113 strain and used sub-inhibitory doses of the two drugs. For nutrient-rich media we used 50 μM 5-FU and 75 μM FUDR, and for nutrient-poor media we used 10 μM 5-FU and 5 μM FUDR. The cultures were grown at 37°C, 200 rpm shaking; diluted 1:200 into fresh medium with/without drug for a total period of 20 days (15 transfers in total). Initially, cultures were diluted every 48 hr to account for the slow growth rate. Last day samples were used for further analysis. We evolved four independent cultures in each condition (drug and media combination). At the end of the evolution experiment the drug-response curve was determined for each independently evolved population in its respective drug and media combination. We then plated evolved populations and isolated 10 individual clones from single colonies. We tested the drug-response curve for these 10 clones and selected a single clone for whole genome sequencing. In most cases, drug resistance was highly similar between individual clones that were sampled from the same population.

## Whole genome sequencing

gDNA was isolated from the chosen evolved clones and from the ancestor strain. For each sample, 0.2 ng of gDNA was used to prepare DNA libraries using Illumina Nextera XT kit according to the kit's instructions. Quality of DNA libraries was assessed using BioAnalyzer/Agilent High Sensitivity DNA Kit. Sequencing was performed using a MiniSeq High Output Reagent Kit, 150 or 300-cycles (Illumina, Cat# FC-420–1002, FC-420–1003). An average of $20\times$ coverage per genome was achieved. Reads were passed through quality-based trimming using quality score cutoff of 30 in Trimmomatic (*Bolger et al., 2014*). Next, reverse and forward reads were merged into a single fastq file. We used the BreSeq tool (*Barrick et al., 2014*) to align the reads to the reference genome (NCBI accession: CP009273) and identify the mutations and genomic rearrangements. We used the gdtools in BreSeq to subtract mutations in the ancestral and control evolved strains from the drug evolved strains. Mutations identified in each evolved population were processed manually into csv spreadsheets and mapped to *E. coli* genomic coordinates for visualization using the Circa software (OMGenomics).

## RNA-sequencing

We grew bacteria overnight in nutrient-poor and nutrient-rich media. Overnight cultures were diluted to $OD_{600}$ of 1 before starting experiments. Cultures were diluted 1:200 into 7 ml of the respective media (without drug) and grown to reach $OD_{600}$ of 0.6. RNA was extracted using Qiagen's RNA-easy protocol with RNA protect (Qiagen, Cat#76526) and RNase-Free DNase (NEB, Cat#M0303S). In these experiments we tested four evolved clones and the ancestral strain using biological triplicates. Library preparation and RNA-sequencing were performed by GeneWiz. The total RNA was depleted from ribosomal RNA and sequenced at 5–10 million reads per sample ($2 \times 150$ bp) on Illumina HiSeq. Paired fastq files per sample were merged and Kallisto (*Bray et al., 2016*) was used to pseudo align RNA-sequencing results to the reference cDNA Ensembl file. Counts from Kallisto were rounded and used as the input for DESeq2 (*Love et al., 2014*), using Wald test, for differential expression analysis. DESeq2 was run with an enrichment cutoff of fourfold and a false-discovery-rate adjusted p-value of 0.1, and a minimum of 10 counts to be included. Enrichment analysis was calculated with GAGE (*Luo et al., 2009*) with a false-discovery-rate adjusted p-value of 0.1.

## Acknowledgements

The research reported in this article was supported by *NIGMS* of the National Institutes of Health under award number R35GM133775 to AM, by DK068429 to AJMW, and by GM122393 to APGG.

## Additional information

### Funding

| Funder | Grant reference number | Author |
|--------|------------------------|--------|
| NIGMS | GM133775 | Amir Mitchell |
| NIGMS | DK068429 | Albertha JM Walhout |
| NIGMS | GM122393 | Aurian P García González |

The funders had no role in study design, data collection and interpretation, or the decision to submit the work for publication.

### Author contributions

Brittany Rosener, Serkan Sayin, Data curation, Software, Formal analysis, Validation, Investigation, Visualization, Methodology, Writing - original draft, Writing - review and editing; Peter O Oluoch, Hirotada Mori, Methodology; Aurian P García González, Data curation, Formal analysis, Funding acquisition, Validation, Investigation, Methodology, Writing - review and editing; Albertha JM Walhout, Formal analysis, Supervision, Funding acquisition, Investigation, Methodology, Writing - original draft, Writing - review and editing; Amir Mitchell, Conceptualization, Data curation, Software, Formal analysis, Supervision, Funding acquisition, Validation, Investigation, Visualization, Methodology, Writing - original draft, Writing - review and editing

### Author ORCIDs

Brittany Rosener (ID) https://orcid.org/0000-0002-1836-8503
Serkan Sayin (ID) https://orcid.org/0000-0001-8776-2240
Peter O Oluoch (ID) http://orcid.org/0000-0001-7451-4993
Hirotada Mori (ID) https://orcid.org/0000-0003-3855-778X
Albertha JM Walhout (ID) http://orcid.org/0000-0001-5587-3608
Amir Mitchell (ID) https://orcid.org/0000-0001-9376-3987

### Decision letter and Author response

Decision letter https://doi.org/10.7554/eLife.59831.sa1
Author response https://doi.org/10.7554/eLife.59831.sa2

## Additional files

### Supplementary files

- Supplementary file 1. Table of barcode frequencies and enrichment across genetic screens.
- Supplementary file 2. Table of gene-set enrichment across genetic screens.
- Supplementary file 3. Table of growth of 43 strains with single gene knockout.
- Supplementary file 4. Table of drug resistance and growth of evolved strains.
- Supplementary file 5. Table of mutations in evolved strains.
- Supplementary file 6. Table of differential gene expression in 5-FU evolved strains.
- Supplementary file 7. Table of gene-set enrichment in 5-FU evolved strains.
- Supplementary file 8. Table of *Caenorhabditis elegans* developmental phenotypes.
- Transparent reporting form

### Data availability

Sequencing data was deposited in SRA under the BioProjects IDs PRJNA645604 and PRJNA645605.

The following datasets were generated:

| Author(s) | Year | Dataset title | Dataset URL | Database and Identifier |
|---|---|---|---|---|
| Rosener B | 2020 | Genome sequencing and transcriptome of 5FU and FUDR evolved Escherichia coli | https://www.ncbi.nlm.nih.gov/bioproject/PRJNA645604 | NCBI BioProject, PRJNA645604 |
| Rosener B | 2020 | E. coli barcoded deletion library screen for antimetabolite chemotherapies 5-FU and FUDR resistance in nutrient-poor & rich media | https://www.ncbi.nlm.nih.gov/bioproject/PRJNA645605 | NCBI BioProject, PRJNA645605 |

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
