## [Decision Letter]

**Acceptance summary:**

Drug treatment can affect the physiology of an organism, but also its microbiome. The response of the microbiome can lead to the modification of its species composition or adaptation of particular ones. This can in turn affect how the host responds to the drug, creating complex dynamics. The authors identify genes and mutations in *E. coli* that confer resistance to chemotherapeutic drugs and show that resistant bacteria can impact the way worm hosts respond to drug treatment. They show that the developmental drug response of worms depends on whether they are fed with resistant bacteria or not. This study provides insight into microbiome-host response dynamics to drug treatment.

**Decision letter after peer review:**

Thank you for submitting your article "Chemotherapy resistance evolved in bacteria lowers drug efficacy in the *Caenorhabditis elegans* host" for consideration by *eLife*. Your article has been reviewed by two peer reviewers, and the evaluation has been overseen by a Reviewing Editor and Gisela Storz as the Senior Editor. The following individual involved in review of your submission has agreed to reveal their identity: Peter Turnbaugh (Reviewer #3).

The reviewers have discussed the reviews with one another and the Reviewing Editor has drafted this decision to help you prepare a revised submission.

Summary:

The authors examine how the evolution of drug resistance in bacteria affects drug toxicity once these bacteria live in a host. They use a powerful combination of *E. coli* functional genomics, experimental evolution and nematode toxicity assays. The work is of interest and the paper is well written and well presented. However, the reviewers raised important points (see below) that would need to be addressed, including better focusing the scope of the manuscript. They would have liked to see more novelty in the results, particularly a better understanding of the mechanisms of resistance at work, for instance fimE mediated resistance. They would have liked as well more in vivo experiments (Reviewer 2) but this would not be required in a revised version as this may necessitate more long term work. The complete list of comments are below to help you improve the manuscript.

Reviewer #1:

Rosener et al. describes a potential new mechanism whereby development of mutations in bacterial genes that confer growth resistance against the inhibitory effects of host targeted drugs indirectly influences the efficacy of these drugs in regulating host physiology. The authors use a barcoding sequencing approach to identify bacterial genes in a single deletion mutant library that confer resistance to growth when challenged with the anticancer drugs 5-FU and FUdR. These were then further tested in an animal model to test whether indirectly they altered the toxicity of these compounds in a worm host model. The authors reassuringly find genes/pathways that have been well and thoroughly described in three previous studies (Garcia-Gonzalez et al., 2017; Scott et al., 2017; Ke, W 2020 Nature Comms). In addition, the authors developed in vitro mutations in *E. coli* against these two drugs in two types of nutritional media (poor and rich) and find that mutations developed in poor media (e.g upp – also found in their genetic-drug screen) confer resistance to the anti-growth effect of 5-FU and FUdR and concomitantly alter the toxicity of these compounds on the host. Importantly, development of resistance in rich media while conferring growth resistance against these compounds does not regulate host toxicity.

Conceptually this is interesting and the experimental work performed is generally sound.

However, the manuscript in its current state lacks novelty and the main claims of development of resistance in bacteria modulating host effects is not substantiated. In fact, their evidence supports the opposite or, at best, that there is a very small chance this is physiologically relevant in mammals/humans.

The authors main statement is that development of bacterial growth resistance against host-targeted drugs leads to a regulation of the toxicity of the drugs in the host indirectly mediated by bacteria.

There are several problems with this claim:

1) The authors often make sweeping generalizations when they have only tested two anti-cancer drugs (chemically related) and one bacterial laboratory strain (*E. coli*). The authors need to be more precise with their language/description which is very vague throughout the entire manuscript and make it clear from the outset what was actually done. It is only half-way through the Introduction that I learned what was actually tested. The Abstract is written deceptively to make the readers believe this is a wide and encompassing study of many drugs etc and that a clearly supported mechanism was found. This is not true and has to be addressed. E.g. anti-metabolites drugs comprise more compounds than 5FU/FUdR…; *E. coli* was the only bacteria used…;

2) The authors have only performed in vitro work to make this claim. The authors often refer to ex-host. This is a meaningless term that the authors use as an attempt to emphasize their work beyond what it really is – in vitro work. This has to be reworded adequately.

3) The authors almost fully dismiss the fact that development of resistance in rich media does not lead to a regulation of the effect of the drug on the host by the microbes which is presented as an unexpected afterthought, when it is anything but. This is essential and actually contradicts their main claim. The mammalian gut is a very rich nutrient environment and growth of a complex microbiota requires nucleotide supplementation (Tramontano et al., 2018). Therefore, developing resistance against 5FU or FUdR in minimal media which does not contain nucleotides, does not reflect the in vivo setting and is physiologically irrelevant. Consequently, development of mutations in bacterial genes (e.g upp) that regulate both growth resistance and mediate the effect of the drug in the host is unlikely to occur in the mammalian gut.

What is striking is that the authors show that bacteria can develop resistance in a more physiologically relevant complex media but these mutations do not lead to regulation of host effects. These findings reflect what can likely occur in the mammalian gut but are mostly ignored to push a message/claim based on data that are unlikely to be physiologically relevant. The genes that are mutated in complex rich media and confer growth resistance against these drugs are novel and it is a shame that this has not been pursued further. Instead, the authors decided to bias their conclusions and focus on genes whose functions and mechanisms are already known.

4) Regardless of the likelihood of one mechanism versus the other to be more physiologically relevant, the burden of proof for the concept being pushed forward to be valid in a mammalian/clinical setting is very low and the authors should therefore be very cautious in their claims. If the authors decide to keep their original message then they have to provide the following to substantiate it:

– They should be able to show that this phenomenon occurs within the gut of mammals treated with the cancer drugs at physiologically relevant concentrations.

– Since the drug load will be shared by the other microbes in the community which will lead to a reduced burden for the microbe of interest the authors should provide evidence that development of resistance occurs in the presence of a complex community of phylogenetically and metabolically diverse microbiota.

– The authors should provide additional evidence that the presence of resistant strains vs sensitive strains or fimE vs upp; is sufficient to regulate host effects in the presence of a complex microbiota.

Overall, the paper is interesting and potentially merits publication if the authors pursue the most novel aspect of their findings (e.g. FimE) and importantly alter/soften their conclusions/claims. The authors have to be more honest and upfront about their work and this needs to be clearly reflected in the title/Abstract and generally throughout the manuscript so that a balanced view of their data is reported. Ultimately, what is shown is that development of *E. coli* resistance to fluoropyrimidine drugs may not lead to toxicity modulation in the host – this should be their title.

Reviewer #3:

Rosener et al. present an analysis of the genetic determinants of bacterial resistance to fluoropyrimidines using complementary experimental evolution and pooled transposon mutagenesis screens. This work adds to the growing literature in this area and brings some much-needed clarity to the overlap between mutations that impact *E. coli* growth and host (*C. elegans*) drug toxicity. The paper is well-written, clear, and concise with a solid body of work.

The one experiment that seemed to be missing is any attempt to replicate either the transposon screen or the experimental evolution experiment in vivo. While I appreciate that this may be technically difficult in worms given that the bacteria are grown in the media (making it hard to distinguish in vitro effects of the drug on the bacterial lawn vs. effect within the GI tract), this data would provide a major advance over the current in vitro results.

[Editors' note: further revisions were suggested prior to acceptance, as described below.]

Thank you for resubmitting your work entitled "Evolved bacterial resistance against fluoropyrimidines can lower chemotherapy impact in the *Caenorhabditis elegans* host" for further consideration by *eLife*. Your revised article has been evaluated by Gisela Storz (Senior Editor) and a Reviewing Editor.

The manuscript has been improved but there are some remaining issues that need to be addressed before acceptance, as outlined below:

Reviewer #1:

The authors present new data regarding the role of fimE in mediating resistance against 5FU. While I appreciate the new data I think these are substandard and the interpretation of the data are incorrect. For example, no drug controls are missing and the curves do not have lag phase. This makes the data hard to interpret both for the calculation of specific growth parameters such as: area under the curve, growth rates and genotype x drug interaction phenotypes. Also, from the data presented, I see that triton rescues the effect of the control strain from 5-FU/FUdr growth inhibition and does NOT abolish the effect of fimE resistance. These specific data need to be dramatically improved and either it conclusively supports the hypothesis that the authors put forward regarding the role provided by fimE mutations in the protective role against these chemotherapeutics or they need to provide additional data that clarifies the role of fimE.

[Editors' note: further revisions were suggested prior to acceptance, as described below.]

Thank you for submitting your article "Evolved bacterial resistance against fluoropyrimidines can lower chemotherapy impact in the *Caenorhabditis elegans* host" for consideration by *eLife*. The evaluation has been overseen by a Reviewing Editor and Gisela Storz as the Senior Editor.

The Reviewing Editor has drafted this decision to help you prepare a revised submission.

Summary:

Drug treatment can affect an organism, but also its microbiome. The response of the microbiome can lead to the modification of its species composition or adaptation of particular species. This can in turn affect how the host responds to the drug. The authors identify genes and mutations in *E. coli* that confer resistance to chemotherapeutic drugs and show that resistant bacteria can impact the way worm hosts respond to drug treatment. They show that the developmental drug response of worms depends on whether they are fed with resistant bacteria or not. This study provides insight into microbiome-host response dynamics to drug treatment.

We were satisfied with the revisions but the new experiments regarding the fimE mutant strain appear to be too preliminary to support the conclusion related to pellicle-like growth is involved in resistance. The authors may resubmit the revised paper that does not include experiments beyond confirming the resistance of this mutant, which we would accept for publication. Preferably, however, the authors would perform additional experiments to substantiate their claims and better work out the details of the molecular mechanisms of resistance. Alternatively, these additional experiments could also be part of future studies if the authors proceeded with the first option.

---

## [Author Response]

Reviewer #1:[…]The authors main statement is that development of bacterial growth resistance against host-targeted drugs leads to a regulation of the toxicity of the drugs in the host indirectly mediated by bacteria.There are several problems with this claim:1) The authors often make sweeping generalizations when they have only tested two anti-cancer drugs (chemically related) and one bacterial laboratory strain (E. coli). The authors need to be more precise with their language/description which is very vague throughout the entire manuscript and make it clear from the outset what was actually done. It is only half-way through the Introduction that I learned what was actually tested. The Abstract is written deceptively to make the readers believe this is a wide and encompassing study of many drugs etc and that a clearly supported mechanism was found. This is not true and has to be addressed. E.g. anti-metabolites drugs comprise more compounds than 5FU/FUdR…; E. coli was the only bacteria used…;

We revised the text in multiple places throughout the manuscript to avoid generalizations and broad claims. Beyond revision of individual sentences, we added new paragraphs in the Discussion section to elaborate on the raised reservations. In order to better clarify the scope of the work as early as possible we changed both the title and the Abstract to indicate we only tested two fluoropyrimidine drugs and used only one bacterial species. We also provide specific details on the work performed as early as possible in the Introduction section (second sentence of the paragraph describing our work).

2) The authors have only performed in vitro work to make this claim. The authors often refer to ex-host. This is a meaningless term that the authors use as an attempt to emphasize their work beyond what it really is – in vitro work. This has to be reworded adequately.

We reworded according to the reviewer request (we discarded the term ex-host and use exclusively in-vitro).

3) The authors almost fully dismiss the fact that development of resistance in rich media does not lead to a regulation of the effect of the drug on the host by the microbes which is presented as an unexpected afterthought, when it is anything but. This is essential and actually contradicts their main claim. The mammalian gut is a very rich nutrient environment and growth of a complex microbiota requires nucleotide supplementation (Tramontano et al., 2018). Therefore, developing resistance against 5FU or FUdR in minimal media which does not contain nucleotides, does not reflect the in vivo setting and is physiologically irrelevant. Consequently, development of mutations in bacterial genes (e.g upp) that regulate both growth resistance and mediate the effect of the drug in the host is unlikely to occur in the mammalian gut.What is striking is that the authors show that bacteria can develop resistance in a more physiologically relevant complex media but these mutations do not lead to regulation of host effects. These findings reflect what can likely occur in the mammalian gut but are mostly ignored to push a message/claim based on data that are unlikely to be physiologically relevant. The genes that are mutated in complex rich media and confer growth resistance against these drugs are novel and it is a shame that this has not been pursued further. Instead, the authors decided to bias their conclusions and focus on genes whose functions and mechanisms are already known.

We thank the reviewer for raising the topic of the host environment. While the conditions in our in-vitro experiments are much removed from the conditions of the host environment, there is clearly a need to include a discussion on nutrient availability in this niche. We added this information to the Discussion section. It is important to note that bacteria interact with their host and with cancer tumors in multiple sites beyond the gut and can therefore create small environments that are influenced by local microbiome-drug-host interactions. While we agree with the reviewer that the gut environment is a nutrient-rich environment, microorganisms are also naturally found in multiple additional sites that may not be nucleotide rich (the skin, mouth, breast ducts, reproductive systems, etc). Tumors formed in these sites will be influenced by the microbiota naturally found in these sites. Additionally, recent works show that solid tumors, even in typically sterile sites (pancreas, liver metastasis), can have persistent bacterial infections. For example, γ-proteobacteria migrating from the gut to pancreatic tumors can create a tumor microenvironment that is impacted by local bacterial metabolism (Geller et al., 2017). Taken the above points into consideration we do not think that we can categorically claim that either nutrient-poor or nutrient-rich media is more representative of the host environment and therefore think that both types of environments merit consideration. We performed all experiments in both media types and always presented the results side-by-side. We revised the Discussion section in order not to bias poor-nutrient or rich-nutrient media and have restated our claims in the Discussion section to be more cautious (claiming that our work only raises the possibility of bacterial selected mutations that impact the host).

4) Regardless of the likelihood of one mechanism versus the other to be more physiologically relevant, the burden of proof for the concept being pushed forward to be valid in a mammalian/clinical setting is very low and the authors should therefore be very cautious in their claims. If the authors decide to keep their original message then they have to provide the following to substantiate it:– They should be able to show that this phenomenon occurs within the gut of mammals treated with the cancer drugs at physiologically relevant concentrations.– Since the drug load will be shared by the other microbes in the community which will lead to a reduced burden for the microbe of interest the authors should provide evidence that development of resistance occurs in the presence of a complex community of phylogenetically and metabolically diverse microbiota.– The authors should provide additional evidence that the presence of resistant strains vs sensitive strains or fimE vs upp; is sufficient to regulate host effects in the presence of a complex microbiota.

We revised the manuscript and present the model system with its inherent limitations and relevance for the mammalian/clinical settings. We revised our claim to state a similar evolutionary principle “may also transpire” in the human setting. As the reviewer requested, we performed additional experiments to study the mechanism underlying resistance by fimE inactivation. The results are added to the manuscript.

Overall, the paper is interesting and potentially merits publication if the authors pursue the most novel aspect of their findings (e.g. FimE) and importantly alter/soften their conclusions/claims. The authors have to be more honest and upfront about their work and this needs to be clearly reflected in the title/Abstract and generally throughout the manuscript so that a balanced view of their data is reported. Ultimately, what is shown is that development of E. coli resistance to fluoropyrimidine drugs may not lead to toxicity modulation in the host – this should be their title.

We thank the reviewer for the detailed comments. We revised the manuscript and specific terminology to provide a balanced view of the results and soften the claims. We revised the title to indicate the host impact is a possibility (not a necessity) of bacterial evolution. We added new results from experiments testing the mechanism underlying drug resistance by inactivation of fimE. Lastly, we revised the Discussion section and added new paragraphs to discuss the relevance of nutrient poor/rich environments to the host niche and the inherent limitations of the model system we used.

Reviewer #3:Rosener et al. present an analysis of the genetic determinants of bacterial resistance to fluoropyrimidines using complementary experimental evolution and pooled transposon mutagenesis screens. This work adds to the growing literature in this area and brings some much-needed clarity to the overlap between mutations that impact E. coli growth and host (C. elegans) drug toxicity. The paper is well-written, clear, and concise with a solid body of work.The one experiment that seemed to be missing is any attempt to replicate either the transposon screen or the experimental evolution experiment in vivo. While I appreciate that this may be technically difficult in worms given that the bacteria are grown in the media (making it hard to distinguish in vitro effects of the drug on the bacterial lawn vs. effect within the GI tract), this data would provide a major advance over the current in vitro results.

We thank the reviewer for the suggestions and comments. We revised the manuscript and addressed these points raised.

[Editors' note: further revisions were suggested prior to acceptance, as described below.]

Reviewer #1:The authors present new data regarding the role of fimE in mediating resistance against 5FU. While I appreciate the new data I think these are substandard and the interpretation of the data are incorrect. For example, no drug controls are missing and the curves do not have lag phase. This makes the data hard to interpret both for the calculation of specific growth parameters such as: area under the curve, growth rates and genotype x drug interaction phenotypes. Also, from the data presented, I see that triton rescues the effect of the control strain from 5-FU/FUdr growth inhibition and does NOT abolish the effect of fimE resistance. These specific data need to be dramatically improved and either it conclusively supports the hypothesis that the authors put forward regarding the role provided by fimE mutations in the protective role against these chemotherapeutics or they need to provide additional data that clarifies the role of fimE.

We thank the reviewer for raising this point. We provide data requested in the revised supplementary figure 5. We further explain our interoperation of the observations in the figure legend. We also agree that addition of no-drug control plots allows to better deconvolve the triton influence and better clarify the rationale behind our conclusions.

1) The data we presented in the growth curves are shown as log absorbance. We did this to allow the reader to identify the logarithmic growth stage in the growth curves (which is a linear on the log scale). The data points corresponding to the lag phase do not show up in the graph since they are beyond the limits of the y-axis (subtracting the basal absorbance resulted in raw values very close the 0 due to low absorbance from the culture). To further clarify the dynamics in the early stages of growth (which include both the lag phase and the early logarithmic growth phase used for our growth rate calculation) we added subplots showing the growth in the first few hours from the same experiment (with absorbance on a linear scale). These subplots allow to clearly observe the lag phase.

2) As requested we now include the no-drug controls as part of the supplementary Figure 5A. As can be observed, addition of triton (without any drug) improves growth of both wild-type and fimE KO strains.

3) Our interpretation is that triton has compound effects in the drug+triton experiment that are strain dependent: it increases growth in both strains (irrespectively of drug) and it counteract resistance only in the fimE KO strain (since it interferes with cell adherence unique to this mutant strain). Since triton by itself effects growth it is misleading to compare the value of the growth rate of a specific strain in drug vs. drug+triton condition. Hence, we propose that comparing growth between the two strains in a specific growth condition is the correct inference strategy.

Here are the underlying details:

In all observed fitness parameters (growth rate during early log, absorbance after 6h, absorbance after 12h, AUC) similar trends emerge: fimE is more resistant than the wild-type in drug while wild-type is equally (or more) resistant than fimE with addition of triton. The only exception we detect is for a short period of time (1-3hr) in 5FU+triton, yet this exception likely arises from a shorter lag phase in the fimE KO. A growth rate calculation for the same period of time (1.5-2.5hr) reveals the two strains have very similar growth rates

In the wild-type strain triton only increases growth (as shown by no-drug experiments). Therefore, in the triton+drug experiment the positive effect of triton on growth is added to the drug effect. This gives the appearance of rescue, yet it is not in fact rescue (since growth improvement is observed even without drug).

In the fimE KO strain triton increases growth (as shown by no-drug experiments) and also counteracts the unique drug resistance mechanism of this strain (cell adherence). Therefore, in the triton+drug experiment triton abolishes resistance by cell adherence and the growth of the KO strain resembles that of the wild-type in the same condition. Here, again, the growth rate is higher in drug+triton than in the drug w/o triton experiment since the positive effect of triton is stronger than its negative effect through interference in cell adherence.

Given that triton influences growth of both strains without drug, it is misleading to directly compare the actual growth rate values of a specific strain in drug against drug+triton conditions (since triton clearly influences growth).

A good strategy to account for the positive effect triton has on growth in both strains is to calculate the growth rate ratio (fimE/wt). Since this division controls for the common positive effect triton has on growth, we can conclude that the remaining effect is strain unique. As supplementary figure 5B shows, this normalization reveals that fimE KO is more drug resistant than the wild-type strain only if the experiment is conducted without triton. This advantage is noticeably abolished with addition of triton.

Given that others have already shown that triton interferes with cell adherence in fimE loss-of-function mutants, we believe that these results indeed support the original hypothesis we put forward – the increased resistance mechanism of the fimE knockout strain against 5FU and FUDR arises from its increased level of cell adherence.

[Editors' note: further revisions were suggested prior to acceptance, as described below.]

Revisions for this paper:We were satisfied with the revisions but the new experiments regarding the fimE mutant strain appear to be too preliminary to support the conclusion related to pellicle-like growth is involved in resistance. The authors may resubmit the revised paper that does not include experiments beyond confirming the resistance of this mutant, which we would accept for publication. Preferably, however, the authors would perform additional experiments to substantiate their claims and better work out the details of the molecular mechanisms of resistance. Alternatively, these additional experiments could also be part of future studies if the authors proceeded with the first option.

We revised the manuscript and figures exactly as suggested. Specifically, we revised the text discussing fimE results and removed any experiments on fimE beyond the ones confirming its increased resistance. We now only mention pellicle-like growth as a possible mechanism for increased resistance and provide reference to works by others. We removed Figure 5—figure supplement 1 from supplementary information.